# Single-Call Stochastic Extragradient Methods for Structured Non-monotone Variational Inequalities: Improved Analysis under Weaker Conditions

**Sayantan Choudhury**
AMS & MINDS
Johns Hopkins University

**Eduard Gorbunov**
MBZUAI

**Nicolas Loizou**
AMS & MINDS
Johns Hopkins University

## Abstract

Single-call stochastic extragradient methods, like stochastic past extragradient (SPEG) and stochastic optimistic gradient (SOG), have gained a lot of interest in recent years and are one of the most efficient algorithms for solving large-scale min-max optimization and variational inequalities problems (VIP) appearing in various machine learning tasks. However, despite their undoubted popularity, current convergence analyses of SPEG and SOG require strong assumptions like bounded variance or growth conditions. In addition, several important questions regarding the convergence properties of these methods are still open, including mini-batching, efficient step-size selection, and convergence guarantees under different sampling strategies. In this work, we address these questions and provide convergence guarantees for two large classes of structured non-monotone VIPs: (i) quasi-strongly monotone problems (a generalization of strongly monotone problems) and (ii) weak Minty variational inequalities (a generalization of monotone and Minty VIPs). We introduce the expected residual condition, explain its benefits, and show how it allows us to obtain a strictly weaker bound than previously used growth conditions, expected co-coercivity, or bounded variance assumptions. Finally, our convergence analysis holds under the arbitrary sampling paradigm, which includes importance sampling and various mini-batching strategies as special cases.

## 1   Introduction

Differentiable game formulations where several parameterized models/players compete to minimize their respective objective functions have recently gained much attention from the machine learning community. Some landmark advances in machine learning that are framed as games (or in their simplified form as min-max optimization problems) are Generative Adversarial Networks (GANs) [19, 2], adversarial training of neural networks [46, 72], reinforcement learning [9, 64], and distributionally robust learning [51, 73].

In this work, we consider a more abstract formulation of the problem and focus on solving the following unconstrained stochastic variational inequality problem (VIP):

$$\text{Find } x^* \in \mathbb{R}^d : \text{ such that } F(x^*) = \frac{1}{n}\sum_{i=1}^{n} F_i(x^*) = 0 \tag{1}$$

where each $F_i : \mathbb{R}^d \to \mathbb{R}^d$ is a Lipschitz continuous operator. Problem (1) generalizes the solution of several types of *stochastic smooth games* [16, 44, 20, 7]. The simplest example is the unconstrained min-max optimization problem (also called a *zero-sum* game):

$$\min_{x_1 \in \mathbb{R}^{d_1}} \max_{x_2 \in \mathbb{R}^{d_2}} \frac{1}{n}\sum_{i=1}^{n} g_i(x_1, x_2), \tag{2}$$

where each component function $g_i : \mathbb{R}^{d_1} \times \mathbb{R}^{d_2} \to \mathbb{R}$ is assumed to be smooth. In this scenario, operator $F_i$ of (1) represents the appropriate concatenation of the block-gradients of $g_i$: $F_i(x) := (\nabla_{x_1} g_i(x_1, x_2); -\nabla_{x_2} g_i(x_1, x_2))$, where $x := (x_1; x_2)$. Solving (1) then amounts to finding a stationary point $x^* = (x_1^*; x_2^*)$ for (2), which under a convex-concavity assumption for $g_i$, implies that it is a global solution for the min-max problem.

However, in modern machine learning applications, game-theoretical formulations that are special cases of problem (1) are rarely monotone. That is, the min-max optimization problem (2) does not satisfy the popular and well-studied convex-concave setting. For this reason, the ML community started focusing on non-monotone problems with extra structural properties.[1] In this work, we focus on such settings (structured non-monotone operators) for which we are able to provide tight convergence guarantees and avoid the standard issues (like cycling and divergence of the methods) appearing in the more general non-monotone regime. In particular, we focus on understanding and efficiently analyze the performance of single-call extragradient methods for solving (i) $\mu$-quasi-strongly monotone VIPs [44, 6] and (ii) weak Minty variational inequalities [14, 33].

**Classes of structured non-monotone VIPs.** Throughout this work we assume that operator $F$ in (1) is $L$- Lipschitz i.e. $\forall x, y \in \mathbb{R}^d$ operator $F$ satisfy $\|F(x) - F(y)\| \leq L\|x - y\|$.

As we have already mentioned, in this work, we deal with two classes of structured non-monotone problems: the $\mu$-quasi strongly monotone VIPs and the weak Minty variational inequalities.

**Definition 1.1.** $F$ is said to be $\mu$-quasi strongly monotone if there is $\mu > 0$ such that:

$$\forall x \in \mathbb{R}^d \qquad \langle F(x), x - x^* \rangle \geq \mu \|x - x^*\|^2. \tag{3}$$

Condition (3) is a relaxation of $\mu$-strong monotonicity, and it includes several non-monotone games as special cases [44]. Inequality (3) can be seen as an extension of the popular quasi-strong convexity assumption from optimization literature [53, 25] to the VIPs [44]. In the literature of variational inequality problems, quasi strongly monotone problems are also known as strong coherent VIPs [66] or VIPs satisfying the strong stability condition [47], or strong Minty variational inequality [14].

One of the weakest possible assumptions on the structure of non-monotone VIPs is the weak Minty variational inequality [14].

**Definition 1.2.** We say weak Minty Variational Inequality (MVI) holds for $F$ if for some $\rho > 0$ :

$$\forall x \in \mathbb{R}^d \qquad \langle F(x), x - x^* \rangle \geq -\rho \|F(x)\|^2. \tag{4}$$

To the best of our knowledge, the weak Minty variational inequality (4) as an assumption was first introduced in [14]. The more popular and extensively studied Minty variational inequality [12, 37, 38, 48] is a particular case of (4) with $\rho = 0$. In addition, the weak MVI condition is implied by the negative comonotonicity [4] or, equivalently, the positive cohypomonotonicity [11]. Finally, when we focus on min-max optimization problems (2), weak MVI condition (with $\rho = 0$) is satisfied for several non-convex non-concave families of min-max objectives, including quasi-convex quasi-concave or star convex- star concave [20]. Extragradient-type methods for solving VIPs satisfying the weak MVI have been proposed in [14, 54] and [8].

## 1.1 Main Contributions
Our main contributions are summarized below.

- **Expected Residual.** We propose the expected residual (ER) condition for stochastic variational inequality problems (1). We explain the benefits of ER and show how it can be used to derive an upper bound on $\mathbb{E}\|g(x)\|^2$ (see Lemma 3.2) that it is strictly weaker than the bounded variance assumption and "growth conditions" previously used for the analysis of stochastic algorithms for solving (1). We prove that ER holds for a large class of operators, i.e., whenever $F_i$ of (1) are Lipschitz continuous.

- **Novel Convergence Guarantees.** We prove the first convergence guarantees for SPEG (7) in the quasi-strongly monotone (3) and weak MVI (4) cases *without using the bounded variance*

---

[1]The computation of approximate first-order locally optimal solutions for general non-monotone problems (without extra structure) is intractable. See [13] and [14] for more details.

Table 1: Summary of known and new convergence results for versions of SEG and SPEG with constant step-sizes applied to solve quasi-strongly monotone variational inequalities and variational inequalities with operators satisfying Weak Minty condition. Columns: "Setup" = quasi-strongly monotone or Weak MVI; "No UBV?" = is the result derived without bounded variance assumption?; "Single-call" = does the method require one oracle call per iteration?; "Convergence rate" = rate of convergence neglecting numerical factors. Notation: $K$ = number of iterations; $L_{\max} = \max_{i \in [n]} L_i$, where $L_i$ is a Lipschitz constant of $F_i$; $\overline{\mu} = \frac{1}{n} \sum_{i=1}^n \mu_i$, where $\mu_i$ is quasi-strong monotonicity constant of $F_i$ (see details in [20]); $\sigma_{\mathrm{US}*}^2 = \frac{1}{n} \sum_{i=1}^n \|F_i(x^*)\|^2$; $\overline{L} = \frac{1}{n} \sum_{i=1}^n L_i$; $\sigma_{\mathrm{IS}*}^2 = \frac{1}{n} \sum_{i=1}^n \frac{\overline{L}}{L_i} \|F_i(x^*)\|^2$; $L$ = Lipschitz constant of $F$; $\mu$ = quasi-strong monotonicity constant of $F$; $\delta, \sigma_*^2$ = parameters from (8); $\rho$ = parameter from Weak Minty condition; $\tau$ = batchsize.

| Setup | Method | No UBV? | Single-call? | Convergence rate |
|---|---|---|---|---|
| Quasi-strong mon. | S-SEG-US [20] | ✓[1] | ✗ | $\frac{L_{\max}}{\overline{\mu}} \exp\left(-\frac{\overline{\mu}}{L_{\max}} K\right) + \frac{\sigma_{\mathrm{US}*}^2}{\overline{\mu}^2 K}$ |
| | S-SEG-IS [20] | ✓[1] | ✗ | $\frac{\overline{L}}{\overline{\mu}} \exp\left(-\frac{\overline{\mu}}{\overline{L}} K\right) + \frac{\sigma_{\mathrm{IS}*}^2}{\overline{\mu}^2 K}$ |
| | SPEG [28] | ✗[2] | ✓ | $\frac{L}{\mu} \exp\left(-\frac{\mu}{L} K\right) + \frac{\sigma_*^2}{\mu^2 K}$ [3] |
| | SPEG (This work) | ✓ | ✓ | $\max\left\{\frac{L}{\mu}, \frac{\delta}{\mu^2}\right\} \exp\left(-\min\left\{\frac{\mu}{L}, \frac{\mu^2}{\delta}\right\} K\right) + \frac{\sigma_*^2}{\mu^2 K}$ |
| Weak MVI[4] | SEG+ [14] | ✗[2] | ✗ | $\frac{L^2 \|x_0 - x^*\|^2}{K(1 - 8\sqrt{2}L\rho)} + \frac{\sigma_*^2}{\tau(1 - 8\sqrt{2}L\rho)}$ [5] |
| | OGDA+ [8] | ✗[2] | ✓ | $\frac{\|x_0 - x^*\|^2}{Kac(a - \rho)} + \frac{\sigma_*^2}{\tau L^2 ac(a - \rho)}$ [6] |
| | SPEG (This work) | ✓ | ✓ | $\frac{\left(1 + \frac{48\omega\gamma\delta}{\tau(1 - L\gamma)^2}\right)^K \|x_0 - x^*\|^2}{K\omega\gamma(1 - L(\gamma + 4\omega))} + \frac{\left(1 + \frac{1 - L\gamma}{K}\left(1 + \frac{48\omega\gamma\delta}{\tau(1 - L\gamma)^2}\right)\right)^K \sigma_*^2}{\tau(1 - L\gamma)(1 - L(\gamma + 4\omega))}$ [7] |

[1] Quasi-strong monotonicity of all $F_i$ is assumed.
[2] It is assumed that (8) holds with $\delta = 0$.
[3] [28] do not derive this result but it can be obtained from their proof using standard choice of step-sizes.
[4] All mentioned results in this case require large batchsizes $\tau = \mathcal{O}(K)$ to get $\mathcal{O}(1/K)$ rate.
[5] The result is derived for $\rho < 1/8\sqrt{2}L$.
[6] The result is derived for $\rho < 3/8L$. Here $a$ and $c$ are assumed to satisfy $aL \leq \frac{7 - \sqrt{1 + 48c^2}}{8(1 + c)}$, $c > 0$ and $a > \rho$.
[7] The result is derived for $\rho < 1/2L$. Here we assume that $\max\{2\rho, 1/(2L)\} < \gamma < 1/L$ and $0 < \omega < \min\{\gamma - 2\rho, (4 - \gamma L)/4L\}$.

*assumption.* We achieve that by using the proposed (ER) condition. In particular, for the class of quasi-strongly monotone VIPs, we show a linear convergence rate to a neighborhood of $x^*$ when constant step-sizes are used. We also provide theoretically motivated step-size switching rules that guarantee exact convergence of SPEG to $x^*$. In the weak MVI case, we prove the convergence of SPEG for $\rho < 1/2L$, improving the existing restrictions on $\rho$. We compare our results with the existing literature in Table 1.

- **Arbitrary Sampling.** Via a stochastic reformulation of the variational inequality problem (1) we explain how our convergence guarantees of SPEG hold under the arbitrary sampling paradigm. This allows us to cover a wide range of samplings for SPEG that were never considered in the literature before, including mini-batching, uniform sampling, and importance sampling as special cases. In this sense, our analysis of SPEG is unified for different sampling strategies. Finally, to highlight the tightness of our analysis, we show that the best-known convergence guarantees of deterministic PEG for strongly monotone and weak MVI can be obtained as special cases of our main theorems.

## 2 Stochastic Reformulation of VIPs & Single-Call Extragradient Methods

In this work, we provide a theoretical analysis of single-call stochastic extragradient methods that allows us to obtain convergence guarantees of any minibatch and reasonable sampling selection. We achieve that by using the recently proposed "stochastic reformulation" of the variational inequality problem (1) from [44]. That is, to allow for any form of minibatching, we use the *arbitrary sampling* notation

$$g(x) = F_v(x) := \frac{1}{n} \sum_{i=1}^n v_i F_i(x), \tag{5}$$

where $v \in \mathbb{R}_+^n$ is a random *sampling vector* drawn from a user-defined distribution $\mathcal{D}$ such that $\mathbb{E}_{\mathcal{D}}[v_i] = 1$, for $i = 1, \ldots, n$. In this setting, the original problem (1) can be equivalently written as,

$$\text{Find } x^* \in \mathbb{R}^d : \mathbb{E}_{\mathcal{D}}\left[F_v(x^*) := \frac{1}{n} \sum_{i=1}^n v_i F_i(x^*)\right] = 0, \tag{6}$$

where the equivalence trivially holds since $\mathbb{E}_{\mathcal{D}}[F_v(x)] = \frac{1}{n} \sum_{i=1}^n \mathbb{E}_{\mathcal{D}}[v_i] F_i(x) = F(x)$.

In this work, we consider *Stochastic Past Extragradient Method* (SPEG) applied to (6):

$$\hat{x}_k = x_k - \gamma_k F_{v_{k-1}}(\hat{x}_{k-1})$$
$$x_{k+1} = x_k - \omega_k F_{v_k}(\hat{x}_k) \tag{7}$$

where $\hat{x}_{-1} = x_0$ and $v^k \sim \mathcal{D}$ is sampled i.i.d at each iteration and $\gamma_k > 0$ and $\omega_k > 0$ are the extrapolation step-size and update step-size respectively. We note that in our convergence analysis, we allow selecting *any* distribution $\mathcal{D}$ that satisfies $\mathbb{E}_{\mathcal{D}}[v_i] = 1 \, \forall i$. This means that for a different selection of $\mathcal{D}$, (7) yields different interpretations of SPEG for solving the original problem (1).

One example of distribution $\mathcal{D}$ is $\tau$–minibatch sampling, which is defined as follows.

**Definition 2.1** ($\tau$-Minibatch sampling). Let $\tau \in [n]$. We say that $v \in \mathbb{R}^n$ is a $\tau$–minibatch sampling if for every subset $S \in [n]$ with $|S| = \tau$, we have that $\mathbb{P}\left[v = \frac{n}{\tau}\sum_{i \in S} e_i\right] := \frac{1}{\binom{n}{\tau}} = \frac{\tau!(n-\tau)!}{n!}$.

By using a double counting argument, one can show that if $v$ is a $\tau$–minibatch sampling, it is also a valid sampling vector ($\mathbb{E}_{\mathcal{D}}[v_i] = 1$) [25]. We highlight that our analysis holds for every form of minibatching and for several choices of sampling vectors $v$. Later in Section 5, we provide more details related to non-uniform sampling. In addition, by Definition 2.1, it is clear that if $\tau = n$, then $v_i = 1$ for all $i \in [n]$. Later in Section 4, we prove how our analysis captures the deterministic Past Extragradient Method as a special case.

In [44], an analysis of stochastic gradient descent-ascent ($x_{k+1} = x_k - \omega_k F_{v_k}(x_k)$) under the arbitrary sampling paradigm was proposed for solving star-co-coercive VIPs. Later [20], extended this approach and provided general convergence guarantees for stochastic extragradient method (SEG) (a stochastic variant of the popular extragradient method [32, 30]) for solving quasi-strongly monotone and monotone VIPs. Despite its popularity, SEG requires two oracle calls per iteration which makes it prohibitively expensive in many large-scale applications and not easily applicable to the online learning problems [18]. This motivates us to explore in detail the convergence guarantees of single-call variants of extragradient methods (extragradient methods that require only a single oracle call per iteration).

**On Single-Call Extragradient Methods.** The seminal work of [56] is the first paper that proposes the deterministic Past Extagradient method. In the stochastic setting, [28] provides an analysis of several stochastic single-call extragradient methods for solving strongly monotone VIPs. In [28], it was also shown that in the unconstrained setting, the update rules of Past Extragradient and Optimistic Gradient are exactly equivalent (see also Proposition B.6 in appendix). Through this connection, and via our stochastic reformulation (6) our theoretical results hold also for the *Stochastic Optimistic Gradient Method* (SOG): $x_{k+1} = x_k - \omega_k F_{v_k}(x_k) - \gamma_k(F_{v_k}(x_k) - F_{v_{k-1}}(x_{k-1}))$. [8] provides the convergence guarantees of SOG for weak MVI. To the best of our knowledge, our work is the first that provides convergence guarantees for SOG under the arbitrary sampling paradigm (captures sampling beyond uniform sampling) and also without using the bounded variance assumption.

## 3 Expected Residual

In our theoretical results, we rely on Expected Residual (ER) condition. In this section, we define ER and explain how it is connected with similar conditions used in optimization literature. We further provide sufficient conditions for ER to hold and prove how it can be used to obtain a strictly weaker upper bound of $\mathbb{E}\|g(x)\|^2$ than previously used growth conditions, expected co-coercivity, or bounded variance assumptions.

**Assumption 3.1.** We say the Expected Residual (ER) condition holds if there is a parameter $\delta > 0$ such that for an unbiased estimator $g(x)$ of the operator $F$, we have

$$\mathbb{E}\left[\|(g(x) - g(x^*)) - (F(x) - F(x^*))\|^2\right] \leq \frac{\delta}{2}\|x - x^*\|^2. \tag{ER}$$

The ER condition bounds how far the stochastic estimator $g(x) = F_v(x)$ (5) used in SPEG is from the true operator $F(x)$. ER depends on both the properties of the operator $F(x)$ and of the selection of sampling (via $g(x)$). Conditions similar to ER appeared before in optimization literature but they

have never been used in operator theory and the analysis of SPEG. In particular, [24] used a similar condition for analyzing SGD in stochastic optimization problems but with the right-hand side of ER to be the function suboptimality $f(x) - f(x^*)$ (such concept is not available in VIPs). In [68] and [22], similar conditions appear under the name "Hessian variance" assumption for distributed minimization problems. In the context of distributed VIPs, a similar but stronger condition to ER is used by [5].

**Bound on Operator Noise.** A common approach for proving the convergence of stochastic algorithms for solving the VIPs is assuming uniform boundedness of the stochastic operator or uniform boundedness of the variance. However, as we explain below, these assumptions either do not hold or are true only for a restrictive set of problems. In our work, we do not assume such bounds. Instead, we use the following direct consequence of ER.

**Lemma 3.2.** Let $\sigma_*^2 := \mathbb{E}\|g(x^*)\|^2 < \infty$ (operator noise at the optimum is finite). If ER holds, then

$$\mathbb{E}\|g(x)\|^2 \le \delta\|x - x^*\|^2 + \|F(x)\|^2 + 2\sigma_*^2. \tag{8}$$

**Sufficient Conditions for ER.** Let us now provide sufficient conditions which guarantee that the ER condition holds and give a closed-form expression for the expected residual parameter $\delta$ and $\sigma_*^2 = \mathbb{E}\|g(x^*)\|^2$ for the case of $\tau$-minibatch sampling (Def. 2.1).

**Proposition 3.3.** Let $F_i$ of problem (1) be $L_i$-Lipschitz operators, then ER holds. If, in addition, vector $v \in \mathbb{R}^n$ is a $\tau$–minibatch sampling (Def. 2.1) then: $\delta = \frac{2}{n\tau}\frac{n-\tau}{n-1}\sum_{i=1}^n L_i^2$, and $\sigma_*^2 = \frac{1}{n\tau}\frac{n-\tau}{n-1}\sum_{i=1}^n \|F_i(x^*)\|^2$.

Similar results to Prop. 3.3 but under different sufficient conditions have been obtained for $\tau$–minibatch sampling under expected smoothness and a variant of expected residual for solving minimization problems in [25] and [24] respectively. In [44], a similar proposition was derived but for the much more restrictive class of co-coercive operators.

**Connection to Other Assumptions.** In the proofs of our convergence results, we use the bound (8), which, as we explained above, is a direct consequence of ER. In this paragraph, we place this bound in a hierarchy of common assumptions used for the analysis of stochastic algorithms for solving VIPs. In the literature on stochastic algorithms for solving the VIPs and min-max optimization problems, previous works assume either bounded operator ($\mathbb{E}\|g(x)\|^2 \le c$) [1, 52], bounded variance ($\mathbb{E}\|g(x) - F(x)\|^2 \le c$) [35, 69, 30] (in Appendix C we provide a simple example where bounded variance assumption does not hold) or growth condition ($\mathbb{E}\|g(x)\|^2 \le c_1\|F(x)\|^2 + c_2$) [36]. In all of these conditions, the parameters $c$, $c_1$, and $c_2$ are usually constants that do not have a closed-form expression. The closer works to our results are [44, 6] which assumes existence of $l_F > 0$ such that the expected co-coercivity condition ($\mathbb{E}\|g(x) - g(x^*)\|^2 \le l_F \langle F(x), x - x^* \rangle$) holds. Their convergence guarantees provide an efficient analysis for several variants of SGDA for solving co-coercive VIPs. In the proposition below, we prove how these conditions are related to the bound (8) obtained using ER.

**Proposition 3.4.** Suppose $F$ is a $L$-Lipschitz operator. Then we have the following hierarchy of assumptions:

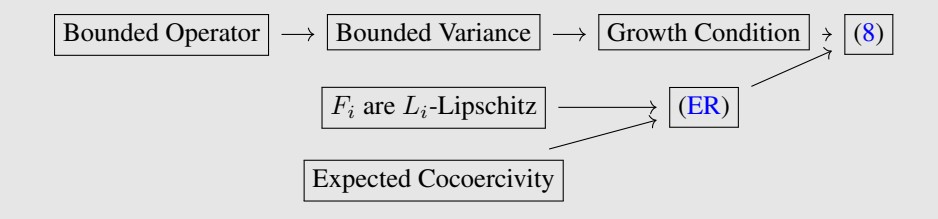

Let us also mention that [29] provided convergence guarantee of double-oracle stochastic extragradient (SEG) method under the variance control condition $\mathbb{E}\|g(x) - F(x)\|^2 \le (a\|x - x^*\| + b)^2$ where $a, b \ge 0$. In their work, they focus on solving VIPs satisfying the error-bound condition, and they did not provide closed-form expressions of parameters $a$ and $b$. Although the analysis of [29]

can be conducted with $a > 0$, the authors only provide rates for the case $a = 0$. The main difference between their results (for SEG) and our results (for SPEG) is that our bound (8) is not really an assumption, but it holds for free when $F_i$ are $L_i$-Lipschitz. In addition, the values of parameters $\delta$ and $\sigma_*^2$ in (8) could have different values based on the sampling used in the update rule of SPEG.

## 4   Convergence Analysis

In this section, we present and discuss the main convergence results of this work. In the first part, we focus on the ones derived for $\mu$-quasi strongly monotone problems (3) (both for constant and decreasing step-sizes), and in the second part on the Weak Minty VIP (4).

### 4.1   Quasi-Strongly Monotone Problems

**Constant Step-size:**   We start with the case of $\mu$-quasi strongly monotone problems and consider the convergence of SPEG with constant step-size.

**Theorem 4.1.** Let $F$ be $L$-Lipschitz, $\mu$-quasi strongly monotone, and let ER hold. Choose step-sizes $\gamma_k = \omega_k = \omega$ such that

$$0 < \omega \leq \min\left\{\frac{\mu}{18\delta}, \frac{1}{4L}\right\} \tag{9}$$

for all $k$. Then the iterates produced by SPEG, given by (7) satisfy

$$R_k^2 \leq \left(1 - \frac{\omega\mu}{2}\right)^k R_0^2 + \frac{24\omega\sigma_*^2}{\mu}, \tag{10}$$

where $R_k^2 := \mathbb{E}\left[\|x_k - x^*\|^2 + \|x_k - \hat{x}_{k-1}\|^2\right]$. Hence, given any $\varepsilon > 0$, and choosing $\omega = \min\left\{\frac{\mu}{18\delta}, \frac{1}{4L}, \frac{\varepsilon\mu}{48\sigma_*^2}\right\}$, SPEG achieves $\mathbb{E}\|x_K - x^*\|^2 \leq \varepsilon$ after $K \geq \max\left\{\frac{8L}{\mu}, \frac{36\delta}{\mu^2}, \frac{96\sigma_*^2}{\varepsilon\mu^2}\right\} \log\left(\frac{2R_0^2}{\varepsilon}\right)$ iterations.

To the best of our knowledge, the above theorem is the first result on the convergence of SPEG that does not rely on the bounded variance assumption. Theorem 4.1 recovers the same rate of convergence with the Independent-Samples SEG (I-SEG) under assumption (8) [20], although [20] simply assume (8), while we show that it follows from Assumption 3.1 holding whenever all summands $F_i$ are Lipschitz. However, in the case when all $F_i$ are $\mu$-quasi strongly monotone and $L_i$-Lipschitz (on average), one can use Same-Sample SEG (S-SEG). The existing results for S-SEG have better exponentially decaying term [49, 20] then Theorem 4.1, e.g., in the case when $L_i = L$ for all $i \in [n]$, we have $\delta = \mathcal{O}(L^2)$ meaning that the exponentially decaying term in (10) is $\mathcal{O}(R_0^2 \exp(-\mu^2 k/L^2)))$, while S-SEG has much better exponentially decaying term $\mathcal{O}(R_0^2 \exp(-\mu k/L)))$.

Such a discrepancy can be partially explained by the following fact: S-SEG can be seen as one step of deterministic Extragradient for stochastic operator $F_{v_k}$ allowing to use one-iteration analysis of Extragradient without controlling the variance. In contrast, there is no version of SPEG that uses the same sample for extrapolation and update steps. This forces to use different samples for these steps and this is a key reason why SPEG cannot be seen as one iteration of deterministic Past-Extragradient for some operator. Due to this, we need to rely on some bound on the variance to handle the stochasticity in the updates; see also [20, Appendix F.1]. Therefore, in our analysis, we use Assumption 3.1, implying (8). Nevertheless, it is still an open question whether it is possible to improve the rate of SPEG in the case of $\mu$-quasi strongly monotone and Lipschitz operators $F_i$.

To highlight the generality of Theorem 4.1, we note that for the deterministic PEG, $\delta = 0$ and $\sigma_*^2 = 0$ (by selecting $\tau = n$ in the definition 2.1 of minibatch sampling). In this case, Theorem 4.1 recovers the well-known result (up to $1/2$ factor in the rate) for deterministic PEG proposed in [17] as shown in the following corollary.

**Corollary 4.2.** Let the assumptions of Theorem 4.1 hold and a deterministic version of SPEG is considered, i.e., $\delta = 0$, $\sigma_*^2 = 0$. Then, Theorem 4.1 implies that for all $k \geq 0$ the iterates produced

by SPEG with step-sizes $\gamma_k = \omega_k = \omega$ such that $0 < \omega \leq \frac{1}{4L}$ satisfy $R_k^2 \leq \left(1 - \frac{\omega\mu}{2}\right)^k R_0^2$, where $R_k^2 := \|x_k - x^*\|^2 + \|x_k - \hat{x}_{k-1}\|^2$.

**Decreasing Step-size:** In this section, we consider two different decreasing step-sizes policies for SPEG applied to solve quasi-strongly monotone problems.

**Theorem 4.3.** Let $F$ be $L$-Lipschitz, $\mu$-quasi strongly monotone, and Assumption 3.1 hold. Let

$$\gamma_k = \omega_k := \begin{cases} \bar{\omega}, & \text{if } k \leq k^*, \\ \frac{2k+1}{(k+1)^2}\frac{2}{\mu}, & \text{if } k > k^*, \end{cases} \tag{11}$$

where $\bar{\omega} := \min\{1/(4L), \mu/(18\delta)\}$ and $k^* = \lceil 4/(\mu\bar{\omega})\rceil$. Then for all $K \geq k^*$ the iterates produced by SPEG with step-sizes (11) satisfy

$$R_K^2 \leq \left(\frac{k^*}{K}\right)^2 \frac{R_0^2}{\exp(2)} + \frac{192\sigma_*^2}{\mu^2 K}, \tag{12}$$

where $R_K^2 := \mathbb{E}\left[\|x_K - x^*\|^2 + \|x_K - \hat{x}_{K-1}\|^2\right]$.

SPEG with step-size policy[2] (11) has two stages of convergence: during first $k^*$ iterations it uses constant step-size to reach some neighborhood of the solution and then the method switches to the decreasing $\mathcal{O}(1/k)$ step-size allowing to reduce the size of the neighborhood.

For the case of strongly monotone problems (a special case of our quasi-strongly monotone setting) [28] also analyze SPEG with decreasing $\mathcal{O}(1/k)$ step-size[3] under bounded variance assumption, i.e., when (8) holds with $\delta = 0$ and some $\sigma_*^2 \geq 0$, which is equivalent to the uniformly bounded variance assumption. In particular, Theorem 5 [28] states $\mathbb{E}\left[\|x_K - x^*\|^2\right] \leq \frac{C\sigma_*^2}{\mu^2 K} + o\left(\frac{1}{K}\right)$ where $C$ is some numerical constant. If the problem is strongly monotone, the result of [28] is closely related to what is obtained in Theorem 4.3: the main difference in the upper-bound is that we provide an explicit form of $o\left(1/K\right)$ term. Moreover, in contrast to the result from [28], Theorem 4.3 holds even when $\delta > 0$ in (8), which covers a larger class of problems.

Following [67, 20, 6], we also consider another decreasing step-size policy.

**Theorem 4.4.** Let $F$ be $L$-Lipschitz, $\mu$-quasi strongly monotone, and Assumption 3.1 hold. Let $\bar{\omega} := \min\{1/(4L), \mu/(18\delta)\}$. If for $K \geq 0$ step-sizes $\{\gamma_k\}_{k\geq 0}$, $\{\omega_k\}_{k\geq 0}$ satisfy $\gamma_k = \omega_k$ and

$$\omega_k := \begin{cases} \bar{\omega}, & \text{if } K \leq \frac{2}{\mu\bar{\omega}}, \\ \bar{\omega}, & \text{if } K > \frac{2}{\mu\bar{\omega}} \text{ and } k \leq k_0, \\ \frac{2}{\frac{2}{\bar{\omega}} + \frac{\mu}{2}(k-k_0)}, & \text{if } K > \frac{2}{\mu\bar{\omega}} \text{ and } k > k_0 \end{cases} \tag{13}$$

where $k_0 = \lceil K/2\rceil$, then the iterates produced by SPEG with the step-sizes defined above satisfy

$$R_K^2 \leq \frac{64R_0^2}{\bar{\omega}\mu}\exp\left\{-\min\left\{\frac{\mu}{16L}, \frac{\mu^2}{72\delta}\right\}K\right\} + \frac{1728\sigma_*^2}{\mu^2 K}, \tag{14}$$

where $R_K^2 := \mathbb{E}\left[\|x_K - x^*\|^2 + \|x_K - \hat{x}_{K-1}\|^2\right]$.

In contrast to (12), the rate from (14) has much better (exponentially decaying) $o\left(1/K\right)$ term. When $\sigma_*^2$ is large and one needs to achieve very good accuracy of the solution, this difference is negligible, since the dominating $\mathcal{O}(1/K)$ term is the same for both bounds (up to numerical factors). However, when $\sigma_*^2$ is small enough, e.g., the model is close to over-parameterized, or it is sufficient to achieve low accuracy of the solution, the dominating term in (14) is typically much smaller than the one from (12). Finally, it is worth mentioning, that the improvement of $o\left(1/K\right)$ is not achieved for free: unlike the policy from (11), step-size rule (13) relies on the knowledge of the total number of steps $K$, which can be inconvenient for the practical use in some cases.

---

[2]Similar step-size policy is used for SGD [25] and SGDA [44].

[3]We point out the proof by [28] can be generalized to the case of constant step-size, though the authors do not consider this step-size schedule explicitly.

## 4.2 Weak Minty Variational Inequality Problems

In this subsection we will discuss convergence of Stochastic Past Extragradient method for Minty Variational Inequality problem. To solve the Minty variational inequality problem we use different step-sizes for SPEG iterates (7).

> **Theorem 4.5.** Let $F$ be $L$-Lipschitz and satisfy Weak Minty condition with parameter $\rho < 1/(2L)$. Let Assumption 3.1 hold. Assume that $\gamma_k = \gamma$, $\omega_k = \omega$ such that $\max\left\{2\rho, \frac{1}{2L}\right\} < \gamma < \frac{1}{L}$ and $0 < \omega < \min\left\{\gamma - 2\rho, \frac{1}{4L} - \frac{\gamma}{4}\right\}$. Then, for all $K \geq 2$ the iterates produced by mini-batched SPEG with batch-size
>
> $$\tau \geq \max\left\{1, \frac{32\delta}{(1-L\gamma)L^3\omega}, \frac{48\omega\gamma\delta(K-1)}{(1-L\gamma)^2}, \frac{2\omega\gamma\sigma_*^2(K-1)}{(1-L\gamma)\|x_0 - x^*\|^2}\right\} \tag{15}$$
>
> satisfy $\min\limits_{0 \leq k \leq K-1} \mathbb{E}\left[\|F(\hat{x}_k)\|^2\right] \leq \frac{C\|x_0 - x^*\|^2}{K-1}$, where $C = \frac{48}{\omega\gamma(1-L(\gamma+4\omega))}$.

The above result establishes $\mathcal{O}(1/K)$ convergence with $\mathcal{O}(K)$ batchsizes for SPEG applied to problems satisfying Weak Minty condition.[4] The closest result is obtained by [8], for the same method under bounded variance assumption, i.e., when $\delta = 0$. In particular, the result of [8] also gives $\mathcal{O}(1/K)$ rate and requires $\mathcal{O}(K)$ batchsizes at each step. We extend this result to the case of non-zeroth $\delta$ and we also improve the assumption on $\rho$: [8] assumes that $\rho < 3/8L$, while Theorem 4.5 holds for $\rho < 1/2L$. The bound on $\rho$ cannot be improved even in the deterministic case [23]. Moreover, it is worth mentioning that the proof of Theorem 4.5 noticeably differs from the one obtained by [8].

In the case of a deterministic oracle, we recover the best-known result for Optimistic Gradient in the Weak Minty setup [8, 23].

> **Corollary 4.6.** Let the assumptions of Theorem 4.5 hold and deterministic version of SPEG is considered, i.e., $\delta = 0$, $\sigma_*^2 = 0$. Then, Theorem 4.5 implies that for all $k \geq 0$ the iterates produced by SPEG with step-sizes $\max\left\{2\rho, \frac{1}{2L}\right\} < \gamma < \frac{1}{L}$ and $0 < \omega < \min\left\{\gamma - 2\rho, \frac{1}{4L} - \frac{\gamma}{4}\right\}$ satisfy
>
> $\min\limits_{0 \leq k \leq K-1} \|F(\hat{x}_k)\|^2 \leq \frac{C\|x_0 - x^*\|^2}{K-1}$, where $C = \frac{48}{\omega\gamma(1-L(\gamma+4\omega))}$.

## 5 Beyond Uniform Sampling

In this section, we illustrate the generality of our analysis by focusing on the non-uniform sampling. In particular, we focus on *single-element sampling* in which only the singleton sets $\{i\}$ for $i = \{1, \ldots, n\}$ have a non-zero probability of being sampled; that is, $\mathbb{P}[|S| = 1] = 1$. We have $\mathbb{P}[v = e_i/p_i] = p_i$. [25] proved that if $v$ is a single-element sampling, it is also a valid sampling vector ($\mathbb{E}_{\mathcal{D}}[v_i] = 1$). With the following proposition, we provide closed-form expressions for the ER parameter $\delta$ and $\sigma_*^2 = \mathbb{E}\|g(x^*)\|^2$ for the case of (non-uniform) single-element sampling.

> **Proposition 5.1.** Let $F_i$ of problem (1) be $L_i$-Lipschitz operators. If, vector $v \in \mathbb{R}^n$ is a single element sampling then $\delta = \frac{2}{n^2}\sum_{i=1}^n \frac{L_i^2}{p_i}$ and $\sigma_*^2 = \frac{1}{n^2}\sum_{i=1}^n \frac{1}{p_i}\|F_i(x^*)\|^2$.

**Importance Sampling.** In importance sampling we aim to choose the probabilities $p_i$ that optimize the iteration complexity. [25] and [20] analyze importance sampling for SGD and SEG respectively. In this work, we provide the first convergence guarantees of SPEG with importance sampling. In particular, we optimize the expected residual parameter $\delta$ with respect to $p_i$, which in turn affects the iteration complexity. Note that, by using Cauchy-Schwarz inequality (20), we have $\sum_{i=1}^n \frac{L_i^2}{p_i} \geq \left(\sum_{i=1}^n L_i\right)^2$, and this lower bound can be achieved for $p_i^\delta = L_i/\sum_{j=1}^n L_j$. In case of importance sampling, we will use these probabilities $p_i^\delta$ which optimizes $\delta$ and define the corresponding $\delta$ as $\delta_{\text{IS}} := \frac{2}{n^2}\left(\sum_{i=1}^n L_i\right)^2$. For uniform sampling $\left(\text{i.e. } p_i = \frac{1}{n}\right)$, the value of the parameter is $\delta_{\text{US}} = \frac{2}{n}\sum_{i=1}^n L_i^2$. Note that, $\delta_{\text{IS}}$ equals $\delta_{\text{US}}$ when all $L_i$ are equal, however $\delta_{\text{IS}}$ can be much smaller than $\delta_{\text{US}}$ when $L_i$ are very different from each other, e.g., when all $L_i$ are relatively small (close to

---

[4]See also Appendix E.5 for a discussion related to the oracle complexity of Theorem 4.5.

zero) and one $L_i$ is large, $\delta_{\text{IS}}$ is almost $n$ times smaller than $\delta_{\text{US}}$. In this latter scenario (when $\delta_{\text{IS}}$ is much smaller than $\delta_{\text{US}}$), importance sampling could be useful and can significantly improve the performance of SPEG. For example, note that the exponentially decaying term in (14) decreases with $\delta$. Hence, this term will decrease much faster with importance sampling than with uniform sampling.

## 6 Numerical Experiments

To verify our theoretical results, we run several experiments on two classes of problems, i.e., strongly monotone problems (a special case of the quasi-strongly monotone VIPs) and weak MVI problems. The code to reproduce our results can be found at https://github.com/isayantan/Single-Call-Stochastic-Extragradient-Methods.

### 6.1 Strongly Monotone Problems

Our experiments consider the quadratic strongly-convex strongly-concave min-max problem from [20]. That is, we implement SPEG on quadratic games of the form $\min_{x \in \mathbb{R}^d} \max_{y \in \mathbb{R}^d} \frac{1}{n} \sum_{i=1}^n f_i(x, y)$ where

$$f_i(x, y) := \frac{1}{2} x^\mathsf{T} A_i x + x^\mathsf{T} B_i y - \frac{1}{2} y^\mathsf{T} C_i y + a_i^\mathsf{T} x - c_i^\mathsf{T} y. \tag{16}$$

Here $A_i, B_i, C_i$ are generated such that the quadratic game is strongly monotone and smooth. In all our experiments, we take $n = 100$ and $d = 30$. We generate positive semi-definite matrices $A_i, B_i, C_i$ such that their eigenvalues lie in the interval $[\mu_A, L_A], [\mu_B, L_B]$ and $[\mu_C, L_C]$ respectively. In all our experiments, we consider $L_A = L_B = L_C = 1$ and $\mu_A = \mu_C = 0.1, \mu_B = 0$ unless otherwise mentioned. The vectors $a_i$ and $c_i$ are generated from $\mathcal{N}_d(0, I_d)$. Here, the $i$th operator is given by

$$F_i \begin{pmatrix} x \\ y \end{pmatrix} = \begin{pmatrix} \nabla_x f_i(x, y) \\ -\nabla_y f_i(x, y) \end{pmatrix} = \begin{pmatrix} A_i x + B_i y + a_i \\ C_i y - B_i^\mathsf{T} x + c_i \end{pmatrix}$$

In Figures 1, 2, and 3, we plot the relative error on the $y$-axis i.e. $\frac{\|x_k - x^*\|^2}{\|x_0 - x^*\|^2}$.

**Constant vs Switching Step-size Rule.** In Fig. 1, we illustrate the step-size switching rule of Theorem 4.3. We place the dotted line to mark when we switch from constant step-size to decreasing step-size. In Fig. 1, the trajectory of switching step-size rule (11) matches that of constant step-size (9) in the first phase (where SPEG runs with constant step-size following (11)). However, it becomes stagnant when the constant step-size SPEG reaches a neighbourhood of optimality. In contrast, the step-size of Theorem 4.3 helps the method to converge to better accuracy.

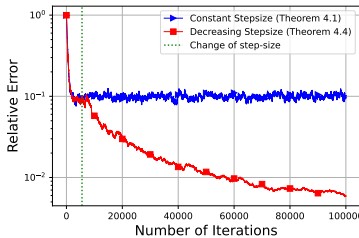

Figure 1: Constant vs Switching

**Comparison to Hsieh et al. [28].** In this experiment, we compare SPEG step-sizes proposed in Theorems 4.1 and 4.3 with step-sizes from [28]. To implement SPEG with the step-sizes from [28], we choose $\gamma$ and $b$ such that $\frac{1}{\mu} < \gamma \leq \frac{b}{4L}$ and set $\omega_k = \gamma_k = \frac{\gamma}{k+b}$. For Fig. 2a, we generate $A_i, B_i, C_i$ as before. First, we sample optimal points $x^*, y^*$ from $\mathcal{N}_d(0, I_d)$ and then generate $a_i, c_i$ such that $F(x^*, y^*) = 0$.

$$\begin{pmatrix} a_i \\ c_i \end{pmatrix} = \begin{pmatrix} A_i & B_i \\ -B_i^\mathsf{T} & C_i \end{pmatrix}^{-1} \begin{pmatrix} x^* \\ y^* \end{pmatrix}.$$

In Fig. 2a, we run the algorithms on interpolated model $(F_i(x^*) = 0$ for all $i \in [n])$. Since the model is interpolated, we have $\sigma_*^2 = 0$ in Theorem 4.1 and linear convergence to the exact optimum asymptotically. In this setting, as shown in Fig. 2a, our proposed step-size results in major improvement compared to the decreasing step-size selection analyzed in [28]. In Fig. 2b, we compare the switching step-size rule with step-size from [28]. In Fig. 2b, we generate $a_i, c_i$ from the normal distribution. In this plot, we manually switch the step-size from constant to decreasing after 305 steps. We observe that such a semi-empirical rule has comparable performance to the step-size selection of Hsieh et al. [28].

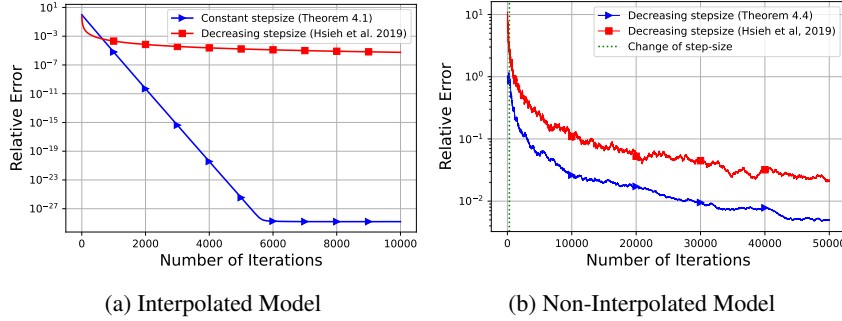

(a) Interpolated Model        (b) Non-Interpolated Model

Figure 2: *Comparison of our* SPEG *using our step-size against decreasing step-size of Hsieh et al. [28]. In plot (a), for constant step-size of* SPEG *we use the upper bound of* (9)*. In plot (b), we run our switching step-size* SPEG (11)*.*

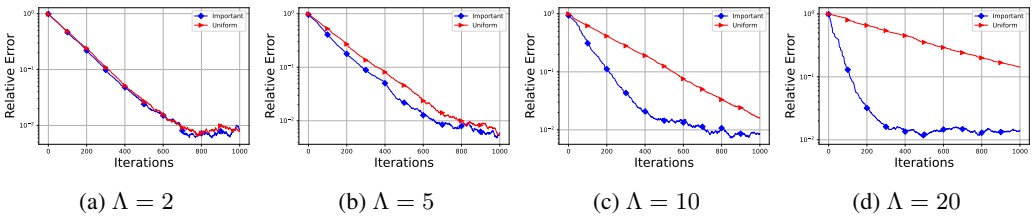

(a) $\Lambda = 2$      (b) $\Lambda = 5$      (c) $\Lambda = 10$      (d) $\Lambda = 20$

Figure 3: *Comparison of* SPEG *with Uniform and Importance Sampling for different* $\Lambda \in \{2, 5, 10, 20\}$*, where the eigenvalues of matrices* $A_1, C_1$ *are uniformly generated from the interval* $[0.1, \Lambda]$*.*

**Uniform vs. Importance Sampling.** In this experiment, we highlight the advantage of using importance sampling over uniform sampling. The eigenvalues of $A_1, C_1$ are uniformly generated from the interval $[0.1, \Lambda]$ while the rest of the matrices are generated as mentioned before. We vary the value of $\Lambda \in \{2, 5, 10, 20\}$ and run and compare SPEG with both uniform and importance sampling (see Fig. 3). For importance sampling, we use the probabilities $p_i = L_i / \sum_{j=1}^n L_j$. In Fig. 3, it is clear that as the value of $\Lambda$ increases, the trajectories under uniform sampling get worse, while the trajectory under importance sampling remains almost identical. This behavior aligns well with our discussion in Section 5.

## 6.2 Weak Minty Variational Inequality Problems

This experiment verifies the convergence guarantees of SPEG in Theorem 4.5. Following the min-max problem mentioned in [8], we consider the objective function

$$\min_{x \in \mathbb{R}} \max_{y \in \mathbb{R}} \frac{1}{n} \sum_{i=1}^n \xi_i xy + \frac{\zeta_i}{2}(x^2 - y^2). \quad (17)$$

In this experiment, we generate $\xi_i, \zeta_i$ such that $L = 8$ and $\rho = 1/32$ for the above min-max problem [8]. We implement SPEG with extrapolation step $\gamma_k = 0.08$ and update step $\omega_k = 0.01$ which satisfies the conditions on step-size in Theorem 4.5. In Fig. 4, we use a batchsize of 6. This plot illustrates that for some weak MVI problems the requirement on the step-size from Theorem 4.5 can be too pessimistic and SPEG with relatively small batchsize achieves reasonable accuracy of the solution. The choice of batchsize ensures that bound (15) holds and $\delta$ is small enough to guarantee convergence of SPEG. We

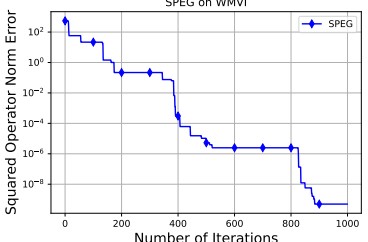

Figure 4: Trajectory of SPEG for solving weak MVI. "Squared Operator Norm Error" in vertical axis denotes the $\min_{0 \le k \le K-1} \mathbb{E}\left[\|F(\hat{x}_k)\|^2\right]$ of Theorem 4.5.

also tried to compare SPEG with SEG+ from [54], however, the authors do not mention their choice of update step-size. We examined several decreasing update step-size for which SEG+ failed to converge. Further details on experiments can be found in Appendix G.1.

## Acknowledgement

Sayantan Choudhury acknowledges support from the Acheson J. Duncan Fund for the Advancement of Research in Statistics. Nicolas Loizou acknowledges support from CISCO Research.

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

# Supplementary Material

We organize the Supplementary Material as follows: Section A discusses the existing literature related to our work. In Section B, we present some technical lemmas required for our analysis, while in Section C, we provide a simple problem where the bounded variance assumption does not hold. Then, in Section D, we provide the proofs of propositions related to Expected Residual. Next, Section E presents the proofs of the main theorems, while a proposition related to arbitrary sampling is proved in Section F. Finally, additional numerical experiments are presented in Section G.

## Contents

# A   Further Related Work

The references necessary to motivate our work and connect it to the most relevant literature are included in the appropriate sections of the main body of the paper. In this section, we present a broader view of the literature, including more details on closely related work and more references to papers that are not directly related to our main results.

- **Classes of Structured Non-monotone Operators.** With an increasing interest in improved computational speed, first-order methods are the primary choice for solving VIPs. However, computation of an approximate first-order locally optimal solution of a general non-monotone VIP is intractable [13, 33]. It motivates us to exploit the additional structures prevalent in large classes of non-monotone VIPs. Recently [20, 28] provide convergence guarantees of stochastic methods for solving quasi-strongly monotone VIPs, while [29] for problems satisfying error-bound conditions. [14] defined the notion of a weak MVI (4) covering classes of non-monotone VIPs.

- **Assumptions on Operator Noise.** The standard analysis of stochastic methods for solving VIPs relies on bounded variance assumption. [8, 14, 28, 17] use bounded variance assumption (i.e. $\mathbb{E}\|F_i(x) - F(x)\|^2 \leq \sigma^2$ for all $x$) while [52, 1] assume bounded operators for their analysis. However, there are examples of simple quadratic games that do not satisfy these conditions. It has motivated researchers to look for alternative/relaxed assumptions on distributions. [44] provides convergence of Stochastic Gradient Descent Ascent Method under Expected Cocoercivity. [29, 49] considered alternative assumptions for analyzing Stochastic Extragradient Methods that do not imply boundedness of the variance. However, there is no analysis of single-call extragradient methods without bounded variance assumption.

- **Weak Minty Variational Inequalities.** Numerous contemporary studies look to identify first-order methods for efficiently solving min-max optimization problems. It varies from simple convex-concave to nontrivial nonconvex nonconcave objectives. Though there has been a significant development in the convex-concave setting, [13] demonstrates that even finding local solutions are intractable for general nonconvex nonconcave objectives. Therefore, researchers seek to identify the structure of objective functions for which it is possible to resolve the intractability issues. [14] proposes the notion of non-monotonicity, which generalizes the existence of a Minty solution (i.e., $\rho = 0$ in (4)). This problem is known as weak Minty variational inequality in the literature. [14, 54] provides convergence guarantees of the Extragradient Method for weak Minty variational inequality. They establish a convergence rate of $\mathcal{O}(1/k)$ for the squared operator norm. [33] shows that it is possible to have an accelerated extragradient method even for non-monotone problems. Furthermore, [8] provides a convergence guarantee for the SOG with a complexity bound of $\mathcal{O}(\varepsilon^{-2})$. However, all papers exploring stochastic extragradient methods for solving weak Minty variational inequality consider bounded variance assumption [8, 14]. Moreover, all algorithms solving Weak Minty variational inequality require increasing batchsize. Recently, [55] introduced BCSEG+ which can solve weak minty variational inequality without increasing batchsize. BCSEG+ involves three oracle calls per iteration and addition of a bias-corrected term in the extrapolation step.

- **Arbitrary Sampling Paradigm.** As we mentioned in the main paper, the stochastic reformulation (6) of the original problem (1) allows us to analyze single-call extragradient methods under the arbitrary sampling paradigm. That is, provide a unified analysis for SPEG that captures multiple sampling strategies, including $\tau$-minibatch and importance samplings. An arbitrary sampling analysis of a stochastic optinmization method was first proposed in the context of the randomized coordinate descent method for solving strongly convex functions in [59]. Since then, several other stochastic methods were studied in this regime, including accelerated coordinate descent algorithms [58, 26], randomized iterative methods for solving consistent linear systems [60, 41, 40], randomized gossip algorithms [39, 42], stochastic gradient descent (SGD) [25, 24], and variance reduced methods [57, 27, 31]. The first analysis of stochastic algorithms under the arbitrary sampling paradigm for solving variational inequality problems was proposed in [43, 44]. In [43, 44], the authors focus on algorithms like the stochastic Hamiltonian method, the stochastic gradient descent ascent, and the stochastic consensus optimization. These ideas were later extended to the case of Stochastic Extragradient by [20]. To the best of our knowledge, our work is the first that provides an analysis of single-call extragradient methods under the arbitrary sampling paradigm.

- **Overparameterized Models and Interpolation.** For a function $f(x) := \frac{1}{n}\sum_{i=1}^{n} f_i(x)$ we say that interpolation condition holds if there exists $x^*$ such that $\min_x f_i(x) = f_i(x^*)$ for all $i \in [n]$ (or equivalently $\nabla f_i(x^*) = 0$ for smooth convex functions) [24]. The interpolation condition is satisfied when the underlying models are sufficiently overparameterized [70]. Some known examples include deep matrix factorization and classification using neural networks [3, 61, 70]. The interpolated model structure enables SGD and other optimization algorithms to have faster convergence [24, 45, 15]. Inspired by this, one can extend the notion of the interpolation condition to operators. In this scenario, we say that the VIP (1) is interpolated if there exists solution $x^*$ of (1) such that $F_i(x^*) = 0$ for all $i \in [n]$. This concept has been explored for analyzing the stochastic extragradient method in [71, 34]. We highlight that our proposed theorems show fast convergence of SPEG in this interpolated regime (when $\sigma_*^2 = 0$). To the best of our knowledge, our work is the first that proves such convergence for SPEG. In Fig. 2a, we experimentally verify the fast convergence for solving a strongly monotone interpolated problem.

- **Deterministic Extragradient Methods.** The Extragradient method (EG) [32] and its single-call variant, Optimistic Gradient (OG) [56], were proposed to overcome the convergence issues of gradient descent-ascent method for solving monotone problems. Since their introduction, these methods have been revisited and explored in various ways. [50] analyzed EG and OG as an approximation of the Proximal Point method to solve bilinear and strongly convex-strongly concave min-max problems. [65] and [62] provide the best-iterate convergence guarantees of EG and OG with a rate of $\mathcal{O}(1/K)$ for solving monotone problems. However, providing a last-iterate convergence rate of EG and OG for monotone VIPs has been a long-lasting open problem that was only recently resolved. The works of [18, 21, 10] prove a last-iterate $\mathcal{O}(1/K)$ convergence rate for these methods. Finally, in the deterministic setting, some recent works provide convergence analysis of EG and OG for solving weak MVI (4) [14, 54, 8, 23].

# B    Technical Preliminaries

Throughout our work, we assume

**Assumption B.1.** Operator $F$ in (1) is L Lipschitz, i.e., $\forall x, y \in \mathbb{R}^d$ operator $F$ satisfies

$$\|F(x) - F(y)\| \leq L\|x - y\|. \tag{18}$$

Operators $F_i : \mathbb{R}^d \to \mathbb{R}^d$ of problem (1) are $L_i$- Lipschitz, i.e., $\forall x, y \in \mathbb{R}^d$ operator $F_i$ satisfies

$$\|F_i(x) - F_i(y)\| \leq L_i\|x - y\|. \tag{19}$$

In our proofs, we often use the following simple inequalities.

**Lemma B.2.** For all $a, b, a_1, a_2, \cdots a_n \in \mathbb{R}^d, n \geq 1, \alpha > 0$, we have the following inequalities:

$$\langle a, b \rangle \leq \|a\|\|b\|, \tag{20}$$

$$\langle a, b \rangle \leq \frac{1}{2\alpha}\|a\|^2 + \frac{\alpha}{2}\|b\|^2, \tag{21}$$

$$\|a + b\|^2 \leq 2\|a\|^2 + 2\|b\|^2, \tag{22}$$

$$\|a\|^2 \geq \frac{1}{2}\|a + b\|^2 - \|b\|^2, \tag{23}$$

$$\left\|\sum_{i=1}^n a_i\right\|^2 \leq n\sum_{i=1}^n \|a_i\|^2. \tag{24}$$

Inequality (22) is well known as Young's Inequality. Now, we present a simple property of unbiased estimators.

**Lemma B.3.** For an unbiased estimator $g$ of operator $F$ i.e. $\mathbb{E}[g(x)] = F(x)$ we have

$$\mathbb{E}\|g(x) - F(x)\|^2 = \mathbb{E}\|g(x)\|^2 - \|F(x)\|^2. \tag{25}$$

Next, we present the following lemma from [67], which plays a vital role in proving the convergence guarantee of Theorem 4.4.

**Lemma B.4.** (Simplified Verison of Lemma 3 from [67]) Let the non-negative sequence $\{r_k\}_{k\geq 0}$ satisfy the relation $r_{k+1} \leq (1 - a\gamma_k)r_k + c\gamma_k^2$ for all $k \geq 0$, parameters $a, c \geq 0$ and any non-negative sequence $\{\gamma_k\}_{k\geq 0}$ such that $\gamma_k \leq \frac{1}{h}$ for some $h \geq a, h > 0$. Then for any $K \geq 0$ one can choose $\{\gamma_k\}_{k\geq 0}$ as follows:

$$\text{if } K \leq \frac{h}{a}, \qquad \gamma_k = \frac{1}{h},$$

$$\text{if } K > \frac{h}{a} \text{ and } k < k_0, \qquad \gamma_k = \frac{1}{h},$$

$$\text{if } K > \frac{h}{a} \text{ and } k \geq k_0, \qquad \gamma_k = \frac{2}{a(\kappa + k - k_0)},$$

where $\kappa = \frac{2h}{a}$ and $k_0 = \lceil\frac{K}{2}\rceil$. For this choice of $\gamma_k$ the following inequality holds:

$$r_K \leq \frac{32hr_0}{a}\exp\left(-\frac{aK}{2h}\right) + \frac{36c}{a^2 K}.$$

We use the next lemma to bound the trace of matrix products.

**Lemma B.5.** For positive semidefinite matrices $A, B \in \mathbb{R}^{d \times d}$ we have

$$\mathrm{tr}(AB) \leq \lambda_{\max}(B)\mathrm{tr}(A), \tag{26}$$

where $\lambda_{\max}(B)$ denotes the maximum eigenvalue of $B$.

Next lemma proves equivalence of SPEG and SOG:

**Proposition B.6** (**Equivalence of SPEG and SOG**). Consider the iterates of SPEG $\{x_k, \hat{x}_k\}_{k=1}^{\infty}$ with constant step-sizes $\omega_k = \omega, \gamma_k = \gamma$ in (7). Then $\hat{x}_k$ follows the iteration rule of SOG i.e.

$$\hat{x}_{k+1} = \hat{x}_k - \omega_k F_{v_k}(\hat{x}_k) - \gamma_k[F_{v_k}(\hat{x}_k) - F_{v_{k-1}}(x_{k-1})] \tag{27}$$

*Proof.* From the update rule of SPEG (7) we get

$$
\begin{aligned}
\hat{x}_{k+1} &= x_{k+1} - \gamma F_{v_k}(\hat{x}_k) \\
&= x_k - \omega F_{v_k}(\hat{x}_k) - \gamma F_{v_k}(\hat{x}_k) \\
&= x_k - (\omega + \gamma) F_{v_k}(\hat{x}_k) \\
&= \hat{x}_k + \gamma F_{v_{k-1}}(\hat{x}_{k-1}) - (\omega + \gamma) F_{v_k}(\hat{x}_k) \\
&= \hat{x}_k - \omega F_{v_k}(\hat{x}_k) - \gamma \Big( F_{v_k}(\hat{x}_k) - F_{v_{k-1}}(\hat{x}_{k-1}) \Big).
\end{aligned}
$$

This shows that SPEG iterations are equivalent to SOG, with $\hat{x}_k$ being the $k$-th iterate of SOG. □

## C   Example: A Problem where the Bounded Variance Condition not Hold

Here, we provide a simple problem that does not satisfy the bounded variance assumption. Consider the linear regression problem

$$\min_{x \in \mathbb{R}} f(x) := \frac{1}{2}(a_1 x - b_1)^2 + \frac{1}{2}(a_2 x - b_2)^2$$

where $x \in \mathbb{R}$. Here $f_1(x) = (a_1 x - b_1)^2$ and $f_2(x) = (a_2 x - b_2)^2$. Now consider the estimator $g(x)$ of $\nabla f(x)$ under uniform sampling i.e. $g(x)$ takes the value $\nabla f_1(x)$ with probability $\frac{1}{2}$ and $\nabla f_2(x)$ with probability $\frac{1}{2}$. Then we have

$$
\begin{aligned}
\mathbb{E}\|g(x) - \nabla f(x)\|^2 &= \frac{1}{2}\|\nabla f_1(x) - \nabla f(x)\|^2 + \frac{1}{2}\|\nabla f_2(x) - \nabla f(x)\|^2 \\
&= \frac{1}{2} \cdot \frac{1}{4}\|\nabla f_1(x) - \nabla f_2(x)\|^2 + \frac{1}{2} \cdot \frac{1}{4}\|\nabla f_2(x) - \nabla f_1(x)\|^2 \\
&= \frac{1}{4}\|\nabla f_1(x) - \nabla f_2(x)\|^2 \\
&= \frac{1}{4}\left(2(a_1 x - b_1)a_1 - 2(a_2 x - b_2)a_2\right)^2 \\
&= \left((a_1^2 - a_2^2)x - (a_1 b_1 - a_2 b_2)\right)^2
\end{aligned}
$$

Therefore, $\mathbb{E}\|g(x) - \nabla f(x)\|^2$ is a quadratic function of $x$ with the coefficient of $x$ being positive. Hence, as $x \to \infty$, we have $\mathbb{E}\|g(x) - \nabla f(x)\|^2 \to \infty$, which means that a constant can not bound the variance.

# D Proofs of Results on Expected Residual

## D.1 Proof of Lemma 3.2

*Proof.* Using Young's Inequality (22), we get

$$\mathbb{E}\|g(x) - F(x)\|^2 \overset{(22)}{\leq} 2\mathbb{E}\|g(x) - F(x) - g(x^*)\|^2 + 2\mathbb{E}\|g(x^*)\|^2$$

$$\overset{(ER)}{\leq} \delta\|x - x^*\|^2 + 2\mathbb{E}\|g(x^*)\|^2.$$

Then breaking down the RHS, we obtain

$$\mathbb{E}\|g(x)\|^2 - \|F(x)\|^2 \overset{(25)}{\leq} \delta\|x - x^*\|^2 + 2\mathbb{E}\|g(x^*)\|^2.$$

Now we rearrange the terms and set $\sigma_*^2 = \mathbb{E}\|g(x^*)\|^2$ to complete the proof of this Lemma. $\qquad\square$

**Proposition D.1.** If $F_i$ are $L_i$-lipschitz then Expected Residual condition (ER) holds. In that case

$$\delta = \frac{2}{n} \sum_{i=1}^n L_i^2 \mathbb{E}(v_i^2).$$

In addition, if $F$ is $\mu$-quasi strongly monotone (3) then we have

$$\delta = \frac{2}{n} \sum_{i=1}^n L_i^2 \mathbb{E}(v_i^2) - 2\mu^2.$$

*Proof.* Note that

$$
\begin{aligned}
\mathbb{E}\|(F_v(x) - F_v(x^*)) - (F(x) - F(x^*))\|^2 &= \mathbb{E}\|F_v(x) - F_v(x^*)\|^2 + \|F(x) - F(x^*)\|^2 \\
&\quad -2\mathbb{E}\langle F_v(x) - F_v(x^*), F(x) - F(x^*)\rangle \\
&= \mathbb{E}\|F_v(x) - F_v(x^*)\|^2 - \|F(x) - F(x^*)\|^2 \\
&= \mathbb{E}\|F_v(x) - F_v(x^*)\|^2 - \|F(x)\|^2 \\
&= \mathbb{E}\left\|\frac{1}{n}\sum_{i=1}^n v_i(F_i(x) - F_i(x^*))\right\|^2 - \|F(x)\|^2 \\
&= \frac{1}{n^2}\mathbb{E}\left\|\sum_{i=1}^n v_i(F_i(x) - F_i(x^*))\right\|^2 - \|F(x)\|^2 \\
&\overset{(24)}{\leq} \frac{1}{n}\sum_{i=1}^n \mathbb{E}(v_i^2)\|F_i(x) - F_i(x^*)\|^2 - \|F(x)\|^2 \\
&\overset{(19)}{\leq} \frac{\|x - x^*\|^2}{n}\sum_{i=1}^n \mathbb{E}(v_i^2)L_i^2 - \|F(x)\|^2. \qquad (28)
\end{aligned}
$$

The first part of the lemma follows by ignoring the positive term $\|F(x)\|^2$. For the second part we assume $F$ is $\mu$-quasi strongly monotone. Then we have

$$\mu\|x - x^*\|^2 \overset{(3)}{\leq} \langle F(x), x - x^*\rangle \overset{(20)}{\leq} \|F(x)\|\|x - x^*\|.$$

Cancelling $\|x - x^*\|$ from both sides we get

$$\mu\|x - x^*\| \leq \|F(x)\|. \qquad (29)$$

Therefore we have the following bound for $\mu$-quasi strongly monotone operator $F$:

$$\mathbb{E}\|(F_v(x) - F_v(x^*)) - (F(x) - F(x^*))\|^2 \overset{(28),(29)}{\leq} \left(\frac{1}{n}\sum_{i=1}^n \mathbb{E}(v_i^2)L_i^2 - \mu^2\right)\|x - x^*\|^2.$$

This proves the second part of the lemma. This lemma ensures that the Lipschitz property is sufficient to guarantee Expected Residual (ER) condition. $\qquad\square$

## D.2 Proof of Proposition 3.3

*Proof.* Proposition D.1 implies that Lipschitzness of all operators $F_i$ is enough to ensure that ER holds. For $\tau$- minibatch sampling, denote the matrix $\mathbf{R} = \left( F_1(x) - F_1(x^*), \cdots, F_n(x) - F_n(x^*) \right) \in \mathbb{R}^{d \times n}$. Then we obtain the following bound:

$$
\begin{aligned}
\mathbb{E}\|F_v(x) - F_v(x^*) - (F(x) - F(x^*))\|^2 &= \mathbb{E}\left\| \frac{1}{n} \sum_{i=1}^{n} v_i(F_i(x) - F_i(x^*)) - (F_i(x) - F_i(x^*)) \right\|^2 \\
&= \frac{1}{n^2} \mathbb{E}\left\| \sum_{i=1}^{n} (v_i - 1)(F_i(x) - F_i(x^*)) \right\|^2 \\
&= \frac{1}{n^2} \mathbb{E}\|\mathbf{R}(v - \mathbf{1})\|^2 \\
&= \frac{1}{n^2} \mathbb{E}(v - \mathbf{1})^\mathsf{T} \mathbf{R}^\mathsf{T} \mathbf{R}(v - \mathbf{1}) \\
&= \frac{1}{n^2} \mathbb{E}\left( \mathrm{tr}\left( \mathbf{R}^\mathsf{T} \mathbf{R}(v - \mathbf{1})(v - \mathbf{1})^\mathsf{T} \right) \right) \\
&= \frac{1}{n^2} \mathrm{tr}\left( \mathbf{R}^\mathsf{T} \mathbf{R} \mathbb{E}\left( (v - \mathbf{1})(v - \mathbf{1})^\mathsf{T} \right) \right) \\
&= \frac{1}{n^2} \mathrm{tr}\left( \mathbf{R}^\mathsf{T} \mathbf{R} \mathbf{Var}[v] \right) \\
&\overset{(26)}{\leq} \frac{\lambda_{\max}(\mathbf{Var}[v])}{n^2} \mathrm{tr}(\mathbf{R}^\mathsf{T} \mathbf{R}) \\
&= \frac{\lambda_{\max}(\mathbf{Var}[v])}{n^2} \sum_{i=1}^{n} \|F_i(x) - F_i(x^*)\|^2 \\
&\overset{(19)}{\leq} \frac{\lambda_{\max}(\mathbf{Var}[v])\|x - x^*\|^2}{n^2} \sum_{i=1}^{n} L_i^2.
\end{aligned}
$$

From the proof details of Lemma F.3 in [63] we have $\lambda_{\max}(\mathbf{Var}[v]) = \frac{n(n-\tau)}{\tau(n-1)}$ for $\tau$-minibatch sampling. Thus we obtain

$$
\mathbb{E}\left\| F_v(x) - F_v(x^*) - (F(x) - F(x^*)) \right\|^2 \leq \frac{2(n-\tau)}{n\tau(n-1)} \sum_{i=1}^{n} L_i^2 \|x - x^*\|^2.
$$

Now we focus on the derivation of $\sigma_*^2 = \mathbb{E}\|F_v(x^*)\|^2$ for $\tau$-minibatch sampling. We expand $\mathbb{E}\|F_v(x^*)\|^2$ as follows:

$$
\begin{aligned}
\mathbb{E}\|F_v(x^*)\|^2 &= \frac{1}{n^2} \mathbb{E}\left\| \sum_{i=1}^{n} v_i F_i(x^*) \right\|^2 \\
&= \frac{1}{n^2} \mathbb{E}\left\| \sum_{i \in S} \frac{1}{p_i} F_i(x^*) \right\|^2 \\
&= \frac{1}{n^2} \mathbb{E}\left\| \sum_{i=1}^{n} \mathbf{1}_{i \in S} \frac{1}{p_i} F_i(x^*) \right\|^2 \\
&= \frac{1}{n^2} \mathbb{E}\left\langle \sum_{i=1}^{n} \mathbf{1}_{i \in S} \frac{1}{p_i} F_i(x^*), \sum_{j=1}^{n} \mathbf{1}_{j \in S} \frac{1}{p_j} F_j(x^*) \right\rangle \\
&= \frac{1}{n^2} \sum_{i,j=1}^{n} \frac{P_{ij}}{p_i p_j} \langle F_i(x^*), F_j(x^*) \rangle, \quad (30)
\end{aligned}
$$

where $P_{ij} = P(i, j \in S)$ and $p_i = P(i \in S)$. For $\tau$-minibatch sampling, we obtain $P_{ij} = \frac{\tau(\tau-1)}{n(n-1)}$ and $p_i = \frac{\tau}{n}$. Plugging in these values of $P_{ij}$ and $p_i$ in (30) we get the closed-form expression of $\sigma_*^2$. This completes the proof of Proposition 3.3. □

### D.3   Proof of Proposition 3.4

Here we enlist the assumptions made on operators. Suppose $g$ is an estimator of operator $F$.

1. **Bounded Operator:**   $\mathbb{E}\|g(x)\|^2 \le \sigma^2$
2. **Bounded Variance:**   $\mathbb{E}\|g(x) - F(x)\|^2 \le \sigma^2$
3. **Growth Condition:**   $\mathbb{E}\|g(x)\|^2 \le \alpha\|F(x)\|^2 + \beta$
4. **Expected Co-coercivity:**   $\mathbb{E}\|g(x) - g(x^*)\|^2 \le l_F \langle F(x), x - x^* \rangle$
5. **Expected Residual:**   $\mathbb{E}\|(g(x) - g(x^*)) - (F(x) - F(x^*))\|^2 \le \dfrac{\delta}{2}\|x - x^*\|^2$
6. **Bound from Lemma 3.2:**   $\mathbb{E}\|g(x)\|^2 \le \delta\|x - x^*\|^2 + \|F(x)\|^2 + 2\sigma_*^2$
7. **$F_i$ are Lipschitz:**   $\|F_i(x) - F_i(y)\| \le L_i\|x - y\|$   $\forall\, i = 1, \dots, n$

*Proof.*  Here we will prove Proposition 3.4

- $1 \implies 2$. Note that $\mathbb{E}\|g(x)\|^2 \le \sigma^2 \le \|F(x)\|^2 + \sigma^2 \implies \mathbb{E}\|g(x) - F(x)\| \le \sigma^2$.

- $2 \implies 3$. Here $\mathbb{E}\|g(x) - F(x)\|^2 \le \sigma^2 \implies \mathbb{E}\|g(x)\|^2 \le \|F(x)\|^2 + \sigma^2$ as $g$ is an unbiased for estimator of $F$. Then take $\alpha = 1$ and $\beta = \sigma^2$.

- $3 \implies 6$. Note that $\mathbb{E}\|g(x)\|^2 \le \alpha\|F(x)\|^2 + \beta \le \alpha L^2\|x - x^*\|^2 + \beta$. The last inequality follows from lipschitz property of $F$ and $F(x^*) = 0$. Then choose $\delta = \alpha L^2$ and $\sigma_*^2 = \beta/2$ to get the result.

- $4 \implies 5$. Note that expected cocoercivity and $L$-Lipschitzness of $F$ imply $\mathbb{E}\|(g(x) - g(x^*)) - (F(x) - F(x^*))\|^2 = \mathbb{E}\|g(x) - g(x^*)\|^2 - \|F(x) - F(x^*)\|^2 \le \mathbb{E}\|g(x) - g(x^*)\|^2 \le l_F \langle F(x), x - x^* \rangle \overset{(B.2)}{\le} \frac{l_F}{2L}\|F(x)\|^2 + \frac{l_F L}{2}\|x - x^*\|^2 \le l_F L\|x - x^*\|^2$.

- $7 \implies 5$. This follows from Proposition D.1.

- $5 \implies 6$. This follows from Lemma 3.2

$\square$

# E   Main Convergence Analysis Results

First, we present some results followed by iterates of SPEG. These will play a key role in proving the Theorems later in this section. Recall that iterates of SPEG are

$$\hat{x}_k = x_k - \gamma_k F_{v_{k-1}}(\hat{x}_{k-1}),$$
$$x_{k+1} = x_k - \omega_k F_{v_k}(\hat{x}_k).$$

**Lemma E.1.** For SPEG iterates with step-size $\omega_k = \gamma_k = \omega$, we have

$$\|x_{k+1} - x^*\|^2 = \|x_{k+1} - \hat{x}_k\|^2 + \|x_k - x^*\|^2 - \|\hat{x}_k - x_k\|^2 - 2\omega \langle F_{v_k}(\hat{x}_k), \hat{x}_k - x^* \rangle. \quad (31)$$

*Proof.* We have

$$
\begin{aligned}
\|x_{k+1} - x^*\|^2 &= \|x_{k+1} - \hat{x}_k + \hat{x}_k - x_k + x_k - x^*\|^2 \\
&= \|x_{k+1} - \hat{x}_k\|^2 + \|\hat{x}_k - x_k\|^2 + \|x_k - x^*\|^2 + 2\langle \hat{x}_k - x_k, x_k - x^* \rangle \\
&\quad +2\langle x_{k+1} - \hat{x}_k, \hat{x}_k - x_k \rangle + 2\langle x_{k+1} - \hat{x}_k, x_k - x^* \rangle \\
&= \|x_{k+1} - \hat{x}_k\|^2 + \|\hat{x}_k - x_k\|^2 + \|x_k - x^*\|^2 + 2\langle x_{k+1} - \hat{x}_k, \hat{x}_k - x^* \rangle \\
&\quad +2\langle \hat{x}_k - x_k, x_k - x^* \rangle \\
&= \|x_{k+1} - \hat{x}_k\|^2 + \|\hat{x}_k - x_k\|^2 + \|x_k - x^*\|^2 + 2\langle x_{k+1} - \hat{x}_k, \hat{x}_k - x^* \rangle \\
&\quad +2\langle \hat{x}_k - x_k, x_k - \hat{x}_k + \hat{x}_k - x^* \rangle \\
&= \|x_{k+1} - \hat{x}_k\|^2 + \|\hat{x}_k - x_k\|^2 + \|x_k - x^*\|^2 + 2\langle x_{k+1} - \hat{x}_k, \hat{x}_k - x^* \rangle \\
&\quad +2\langle \hat{x}_k - x_k, \hat{x}_k - x^* \rangle - 2\|\hat{x}_k - x_k\|^2 \\
&= \|x_{k+1} - \hat{x}_k\|^2 - \|\hat{x}_k - x_k\|^2 + \|x_k - x^*\|^2 + 2\langle x_{k+1} - \hat{x}_k, \hat{x}_k - x^* \rangle \\
&\quad +2\langle \hat{x}_k - x_k, \hat{x}_k - x^* \rangle \\
&= \|x_{k+1} - \hat{x}_k\|^2 - \|\hat{x}_k - x_k\|^2 + \|x_k - x^*\|^2 + 2\langle x_{k+1} - x_k, \hat{x}_k - x^* \rangle \\
&= \|x_{k+1} - \hat{x}_k\|^2 - \|\hat{x}_k - x_k\|^2 + \|x_k - x^*\|^2 - 2\omega \langle F_{v_k}(\hat{x}_k), \hat{x}_k - x^* \rangle.
\end{aligned}
$$

$\square$

**Lemma E.2.** Let $F$ be $L$-Lipschitz, and let ER hold. Then SPEG iterates satisfy

$$\mathbb{E}_{\mathcal{D}}\|F_{v_k}(\hat{x}_k) - F_{v_{k-1}}(\hat{x}_{k-1})\|^2 \leq \delta\|\hat{x}_k - x^*\|^2 + 2\delta\|\hat{x}_{k-1} - x^*\|^2 + 2L^2\|\hat{x}_k - \hat{x}_{k-1}\|^2 + 6\sigma_*^2. \quad (32)$$

*Proof.*

$$
\begin{aligned}
\mathbb{E}_{\mathcal{D}}\|F_{v_k}(\hat{x}_k) - F_{v_{k-1}}(\hat{x}_{k-1})\|^2 &= \mathbb{E}_{\mathcal{D}}\|F_{v_k}(\hat{x}_k) - F(\hat{x}_k)\|^2 + \mathbb{E}_{\mathcal{D}}\|F(\hat{x}_k) - F_{v_{k-1}}(\hat{x}_{k-1})\|^2 \\
&\quad +2\mathbb{E}_{\mathcal{D}}\langle F_{v_k}(\hat{x}_k) - F(\hat{x}_k), F(\hat{x}_k) - F_{v_{k-1}}(\hat{x}_{k-1}) \rangle \\
&= \mathbb{E}_{v_k}\|F_{v_k}(\hat{x}_k) - F(\hat{x}_k)\|^2 + \mathbb{E}_{\mathcal{D}}\|F(\hat{x}_k) - F_{v_{k-1}}(\hat{x}_{k-1})\|^2 \\
&\overset{(22)}{\leq} \mathbb{E}_{\mathcal{D}}\|F_{v_k}(\hat{x}_k) - F(\hat{x}_k)\|^2 + 2\mathbb{E}_{\mathcal{D}}\|F(\hat{x}_k) - F(\hat{x}_{k-1})\|^2 \\
&\quad +2\mathbb{E}_{\mathcal{D}}\|F(\hat{x}_{k-1}) - F_{v_{k-1}}(\hat{x}_{k-1})\|^2 \\
&= \mathbb{E}_{\mathcal{D}}\|F_{v_k}(\hat{x}_k)\|^2 - \|F(\hat{x}_k)\|^2 + 2\|F(\hat{x}_k) - F(\hat{x}_{k-1})\|^2 \\
&\quad +2\mathbb{E}_{\mathcal{D}}\|F_{v_{k-1}}(\hat{x}_{k-1})\|^2 - 2\|F(\hat{x}_{k-1})\|^2 \\
&\overset{(8)}{\leq} \delta\|\hat{x}_k - x^*\|^2 + 2\delta\|\hat{x}_{k-1} - x^*\|^2 + 6\sigma_*^2 \\
&\quad +2\|F(\hat{x}_k) - F(\hat{x}_{k-1})\|^2 \\
&\overset{(18)}{\leq} \delta\|\hat{x}_k - x^*\|^2 + 2\delta\|\hat{x}_{k-1} - x^*\|^2 + 6\sigma_*^2 \\
&\quad +2L^2\|\hat{x}_k - \hat{x}_{k-1}\|^2.
\end{aligned}
$$

$\square$

**Lemma E.3.** For $\omega \in \left[0, \frac{1}{4L}\right]$ the following two conditions hold:

$$2\omega(\mu - \omega\delta) + 8\omega^2 L^2 - 1 \le 0, \tag{33}$$

$$\text{and} \quad 8\omega^2(\delta + L^2) \le 1 - \omega\mu + 9\omega^2\delta. \tag{34}$$

*Proof.* Note that for $\omega \in \left[0, \frac{1}{4L}\right]$, we have

$$2\omega(\mu - \omega\delta) + 8\omega^2 L^2 - 1 \overset{\omega^2\delta \ge 0}{\le} 2\omega\mu + 8\omega^2 L^2 - 1 \overset{\omega \le \frac{1}{4L}}{\le} \frac{\mu}{2L} + \frac{1}{2} - 1 \overset{\mu \le L}{\le} 0.$$

This proves the first condition. The second condition is equivalent to $\omega(\mu - \omega\delta) + 8\omega^2 L^2 - 1 \le 0$, which is again true using similar arguments. $\quad\square$

### E.1 Proof of Theorem 4.1

*Proof.* For $\omega \in \left[0, \frac{\mu}{18\delta}\right]$ we have $\omega(\mu - 9\omega\delta) \ge 0$ and $1 - \omega(\mu - 9\omega\delta) \le 1 - \frac{\omega\mu}{2}$. Then we derive

$$
\begin{aligned}
\mathbb{E}_{\mathcal{D}}[\|x_{k+1} - x^*\|^2 + \|x_{k+1} - \hat{x}_k\|^2] \overset{(31)}{=} & \ \|x_k - x^*\|^2 + 2\mathbb{E}_{\mathcal{D}}\|x_{k+1} - \hat{x}_k\|^2 - \|\hat{x}_k - x_k\|^2 \\
& -2\omega\mathbb{E}_{\mathcal{D}}\langle F_{v_k}(\hat{x}_k), \hat{x}_k - x^*\rangle \\
= & \ \|x_k - x^*\|^2 + 2\mathbb{E}_{\mathcal{D}}\|x_{k+1} - \hat{x}_k\|^2 - \|\hat{x}_k - x_k\|^2 \\
& -2\omega\langle F(\hat{x}_k), \hat{x}_k - x^*\rangle \\
\overset{(3)}{\le} & \ \|x_k - x^*\|^2 + 2\mathbb{E}_{\mathcal{D}}\|x_{k+1} - \hat{x}_k\|^2 - \|\hat{x}_k - x_k\|^2 \\
& -2\omega\mu\|\hat{x}_k - x^*\|^2 \\
= & \ \|x_k - x^*\|^2 + 2\omega^2\mathbb{E}_{\mathcal{D}}\|F_{v_k}(\hat{x}_k) - F_{v_{k-1}}(\hat{x}_{k-1})\|^2 \\
& -\|\hat{x}_k - x_k\|^2 - 2\omega\mu\|\hat{x}_k - x^*\|^2 \\
\overset{(32)}{\le} & \ \|x_k - x^*\|^2 + 2\omega^2\Big(\delta\|\hat{x}_k - x^*\|^2 + 2\delta\|\hat{x}_{k-1} - x^*\|^2 \\
& +2L^2\|\hat{x}_k - \hat{x}_{k-1}\|^2 + 6\sigma_*^2\Big) - \|\hat{x}_k - x_k\|^2 \\
& -2\omega\mu\|\hat{x}_k - x^*\|^2 \\
= & \ \|x_k - x^*\|^2 - 2\omega(\mu - \omega\delta)\|\hat{x}_k - x^*\|^2 \\
& +4\omega^2\delta\|\hat{x}_{k-1} - x^*\|^2 + 4\omega^2 L^2\|\hat{x}_k - \hat{x}_{k-1}\|^2 \\
& -\|\hat{x}_k - x_k\|^2 + 12\omega^2\sigma_*^2 \\
\overset{(22)}{\le} & \ \|x_k - x^*\|^2 - \omega(\mu - \omega\delta)\|x_k - x^*\|^2 \\
& +2\omega(\mu - \omega\delta)\|x_k - \hat{x}_k\|^2 + 4\omega^2\delta\|\hat{x}_{k-1} - x^*\|^2 \\
& +4\omega^2 L^2\|\hat{x}_k - \hat{x}_{k-1}\|^2 - \|\hat{x}_k - x_k\|^2 \\
& +12\omega^2\sigma_*^2 \\
\overset{(22)}{\le} & \ \|x_k - x^*\|^2 - \omega(\mu - \omega\delta)\|x_k - x^*\|^2 \\
& +2\omega(\mu - \omega\delta)\|x_k - \hat{x}_k\|^2 + 8\omega^2\delta\|\hat{x}_{k-1} - x_k\|^2 \\
& +8\omega^2\delta\|x_k - x^*\|^2 + 8\omega^2 L^2\|\hat{x}_k - x_k\|^2 \\
& +8\omega^2 L^2\|x_k - \hat{x}_{k-1}\|^2 - \|\hat{x}_k - x_k\|^2 + 12\omega^2\sigma_*^2 \\
= & \ (1 - \omega\mu + 9\omega^2\delta)\|x_k - x^*\|^2 \\
& +(8\omega^2\delta + 8\omega^2 L^2)\|x_k - \hat{x}_{k-1}\|^2 \\
& +(2\omega(\mu - \omega\delta) + 8\omega^2 L^2 - 1)\|x_k - \hat{x}_k\|^2 + 12\omega^2\sigma_*^2.
\end{aligned}
$$

Then using (33), (34) we have

$$\mathbb{E}_{\mathcal{D}}[\|x_{k+1} - x^*\|^2 + \|x_{k+1} - \hat{x}_k\|^2] \leq (1 - \omega\mu + 9\omega^2\delta)\left(\|x_k - x^*\|^2 + \|x_k - \hat{x}_{k-1}\|^2\right)$$
$$+ 12\omega^2\sigma_*^2.$$

Then we take total expectation with respect to the algorithm to obtain the following recurrence:

$$R_{k+1}^2 \leq (1 - \omega\mu + 9\omega^2\delta)R_k^2 + 12\omega^2\sigma_*^2. \tag{35}$$

Using the inequality $1 - \omega(\mu - 9\omega\delta) \leq 1 - \frac{\omega\mu}{2}$, we have

$$\mathbb{E}\left[\|x_{k+1} - x^*\|^2 + \|x_{k+1} - \hat{x}_k\|^2\right] \leq \left(1 - \frac{\omega\mu}{2}\right)\mathbb{E}\left[\|x_k - x^*\|^2 + \|x_k - \hat{x}_{k-1}\|^2\right] + 12\omega^2\sigma_*^2. \tag{36}$$

The theorem follows by unrolling the above recurrence. In order to compute the iteration complexity of SPEG, we consider any arbitrary $\varepsilon > 0$. Then we choose the step-size $\omega$ such that $\frac{24\omega\sigma_*^2}{\mu} \leq \frac{\varepsilon}{2}$ i.e. $\omega \leq \frac{\varepsilon\mu}{48\sigma_*^2}$. Next we will choose the number of iterations $k$ such that $(1 - \frac{\omega\mu}{2})^k R_0^2 \leq \frac{\varepsilon}{2}$. It is equivalent to choosing $k$ such that

$$\log\left(\frac{2R_0^2}{\varepsilon}\right) \leq k\log\left(\frac{1}{1 - \frac{\omega\mu}{2}}\right).$$

Now using the fact $\log\left(\frac{1}{\rho}\right) \geq 1 - \rho$ for $0 < \rho \leq 1$, we get $\log\left(\frac{2R_0^2}{\varepsilon}\right) \leq \frac{k\omega\mu}{2}$, or equivalently $k \geq \frac{2}{\omega\mu}\log\left(\frac{2R_0^2}{\varepsilon}\right)$. Therefore, with step-size $\omega = \min\left\{\frac{\mu}{18\delta}, \frac{1}{4L}, \frac{\varepsilon\mu}{48\sigma_*^2}\right\}$ we get the following lower bound on the number of iterations

$$k \geq \max\left\{\frac{8L}{\mu}, \frac{36\delta}{\mu^2}, \frac{96\sigma_*^2}{\varepsilon\mu^2}\right\}\log\left(\frac{2R_0^2}{\varepsilon}\right).$$

$\square$

### E.2 Proof of Theorem 4.3

*Proof.* For $\omega \leq \min\left\{\frac{1}{4L}, \frac{\mu}{18\delta}\right\}$, from Theorem 4.1 we obtain

$$R_{k+1}^2 \leq \left(1 - \frac{\omega\mu}{2}\right)^{k+1} R_0^2 + \frac{24\omega\sigma_*^2}{\mu}.$$

Let the step-size $\omega_k = \frac{2k+1}{(k+1)^2}\frac{2}{\mu}$ and $k^*$ be an integer that satisfies $\omega_{k^*} \leq \bar{\omega}$. In particular this holds when $k^* \geq \left\lceil\frac{4}{\mu\bar{\omega}} - 1\right\rceil$. Note that $\omega_k$ is decreasing in $k$ and consequently $\omega_k \leq \bar{\omega}$ for all $k \geq k^*$. Therefore, from (36) we derive

$$R_{k+1}^2 \leq \left(1 - \omega_k\frac{\mu}{2}\right)R_k^2 + 12\omega_k^2\sigma_*^2$$

for all $k \geq k^*$. Then we replace $\omega_k$ with $\frac{2k+1}{(k+1)^2}\frac{2}{\mu}$ to obtain

$$R_{k+1}^2 \leq \left(1 - \frac{2k+1}{(k+1)^2}\right)R_k^2 + 48\sigma_*^2\frac{(2k+1)^2}{\mu^2(k+1)^4}$$
$$= \frac{k^2}{(k+1)^2}R_k^2 + 48\sigma_*^2\frac{(2k+1)^2}{\mu^2(k+1)^4}.$$

Multiplying both sides by $(k+1)^2$ we get

$$(k+1)^2 R_{k+1}^2 \leq k^2 R_k^2 + \frac{48\sigma_*^2}{\mu^2}\left(\frac{2k+1}{k+1}\right)^2$$
$$\leq k^2 R_k^2 + \frac{192\sigma_*^2}{\mu^2},$$

where in the last line follows from $\frac{2k+1}{k+1} < 2$. Rearranging and summing the last expression for $t = k^*, \cdots, k$ we obtain

$$\sum_{t=k^*}^{k} (t+1)^2 R_{t+1}^2 - t^2 R_t^2 \leq \frac{192\sigma_*^2}{\mu^2}(k - k^*).$$

Using telescopic sum and dividing both sides by $(k + 1)^2$ we obtain

$$R_{k+1}^2 \leq \left(\frac{k^*}{k+1}\right)^2 R_{k^*}^2 + \frac{192\sigma_*^2(k-k^*)}{\mu^2(k+1)^2}. \tag{37}$$

Suppose for $k \leq k^*$, we have $\omega_k = \bar{\omega} = \min\left\{\frac{1}{4L}, \frac{\mu}{18\delta}\right\}$ i.e. constant step-size. Then from (10), we obtain $R_{k^*}^2 \leq \left(1 - \frac{\mu\bar{\omega}}{2}\right)^{k^*} R_0^2 + \frac{24\bar{\omega}\sigma_*^2}{\mu}$. This bound on $R_{k^*}^2$, combined with (37) yields

$$R_{k+1}^2 \leq \left(\frac{k^*}{k+1}\right)^2 \left(1 - \frac{\mu\bar{\omega}}{2}\right)^{k^*} R_0^2 + \left(\frac{k^*}{k+1}\right)^2 \frac{24\bar{\omega}\sigma_*^2}{\mu} + \frac{192\sigma_*^2(k-k^*)}{\mu^2(k+1)^2}.$$

Now we want to choose $k^*$ which minimizes the expression $\left(\frac{k^*}{k+1}\right)^2 \frac{24\bar{\omega}\sigma_*^2}{\mu} + \frac{192\sigma_*^2(k-k^*)}{\mu^2(k+1)^2}$. Note that, it is minimized at $\frac{4}{\mu\bar{\omega}}$, hence we choose $k^* = \left\lceil \frac{4}{\mu\bar{\omega}} \right\rceil$. Therefore, using this value of $k^*$, we obtain

$$
\begin{aligned}
R_{k+1}^2 &\leq \left(\frac{k^*}{k+1}\right)^2 \left(1 - \frac{2}{k^*}\right)^{k^*} R_0^2 + \frac{24\sigma_*^2}{\mu^2(k+1)^2}(8k - 4k^*) \\
&\leq \left(\frac{k^*}{k+1}\right)^2 \left(1 - \frac{2}{k^*}\right)^{k^*} R_0^2 + \frac{192k\sigma_*^2}{\mu^2(k+1)^2} \\
&\leq \left(\frac{k^*}{k+1}\right)^2 \frac{R_0^2}{e^2} + \frac{192\sigma_*^2}{\mu^2(k+1)}.
\end{aligned}
$$

The last line follows from $\left(1 - \frac{1}{x}\right)^x \leq e^{-1}$ for all $x \geq 1$. This completes the proof. $\qquad\square$

### E.3  Proof of Theorem 4.4

*Proof.* For $0 < \omega_k \leq \left\{\frac{1}{4L}, \frac{\mu}{18\delta}\right\}$ we obtain the following bound from Theorem 4.1:

$$R_k^2 \leq \left(1 - \frac{\mu\omega_k}{2}\right) R_{k-1}^2 + 12\omega_k^2\sigma_*^2.$$

Then using Lemma B.4 with $a = \frac{\mu}{2}, h = \frac{1}{\omega}$ and $c = 12\sigma_*^2$ we complete the proof of this Theorem. $\quad\square$

### E.4  Proof of Theorem E.4

**Theorem E.4.** Let $F$ be $L$-Lipschitz and satisfy Weak Minty condition with parameter $\rho < 1/(2L)$. Assume that inequality (8) holds (e.g., it holds whenever Assumption 3.1 holds, see Lemma 3.2). Assume that $\gamma_k = \gamma, \omega_k = \omega$ and

$$\max\left\{2\rho, \frac{1}{2L}\right\} < \gamma < \frac{1}{L}, \quad 0 < \omega < \min\left\{\gamma - 2\rho, \frac{1}{4L} - \frac{\gamma}{4}\right\}, \quad \delta \leq \frac{(1-L\gamma)L^3\omega}{32}.$$

Then, for all $K \geq 2$ the iterates produced by SPEG satisfy

$$
\min_{0 \leq k \leq K-1} \mathbb{E}\left[\|F(\hat{x}_k)\|^2\right] \leq \frac{\left(1 + 8\omega\gamma(\delta + L^2) - L\gamma\right)\left(1 + \frac{48\omega\gamma\delta}{(1-L\gamma)^2}\right)^{K-1}\|x_0 - x^*\|^2}{\omega\gamma(1 - L(\gamma + 4\omega))(K-1)}
$$
$$
+ \frac{8\left(8 + \frac{(1-L\gamma)^2}{K-1}\left(1 + \frac{48\omega\gamma\delta}{(1-L\gamma)^2}\right)^{K-1}\right)\sigma_*^2}{(1 - L\gamma)^2(1 - L(\gamma + 4\omega))}. \tag{38}
$$

*Proof.* The proof closely follows the proof of Lemma C.3 and Theorem C.4 from [23]. The update rule of SPEG implies for $k \geq 1$

$$
\begin{aligned}
\|x_{k+1} - x^*\|^2 &= \|x_k - x^*\|^2 - 2\omega\langle x_k - x^*, F_{v_k}(\hat{x}_k)\rangle + \omega^2\|F_{v_k}(\hat{x}_k)\|^2 \\
&= \|x_k - x^*\|^2 - 2\omega\langle \hat{x}_k - x^*, F_{v_k}(\hat{x}_k)\rangle - 2\omega\gamma\langle F_{v_{k-1}}(\hat{x}_{k-1}), F_{v_k}(\hat{x}_k)\rangle \\
&\quad + \omega^2\|F_{v_k}(\hat{x}_k)\|^2 \\
&= \|x_k - x^*\|^2 - 2\omega\langle \hat{x}_k - x^*, F_{v_k}(\hat{x}_k)\rangle - \omega\gamma\|F_{v_{k-1}}(\hat{x}_{k-1})\|^2 \\
&\quad - \omega(\gamma - \omega)\|F_{v_k}(\hat{x}_k)\|^2 + \omega\gamma\|F_{v_k}(\hat{x}_k) - F_{v_{k-1}}(\hat{x}_{k-1})\|^2,
\end{aligned}
$$

where in the last step we apply $2\langle a, b\rangle = \|a\|^2 + \|b\|^2 - \|a - b\|^2$, which holds for all $a, b \in \mathbb{R}^d$. Taking the full expectation and using $\mathbb{E}[\mathbb{E}_{v_k}[\cdot]] = \mathbb{E}[\cdot]$ and Weak Minty condition, we derive

$$
\begin{aligned}
\mathbb{E}\left[\|x_{k+1} - x^*\|^2\right] &\leq \mathbb{E}\left[\|x_k - x^*\|^2\right] - 2\omega\mathbb{E}\left[\langle \hat{x}_k - x^*, F(\hat{x}_k)\rangle\right] - \omega\gamma\mathbb{E}\left[\|F_{v_{k-1}}(\hat{x}_{k-1})\|^2\right] \\
&\quad - \omega(\gamma - \omega)\mathbb{E}\left[\|F_{v_k}(\hat{x}_k)\|^2\right] + \omega\gamma\mathbb{E}\left[\|F_{v_k}(\hat{x}_k) - F_{v_{k-1}}(\hat{x}_{k-1})\|^2\right] \\
&\overset{(4)}{\leq} \mathbb{E}\left[\|x_k - x^*\|^2\right] + 2\omega\rho\mathbb{E}\left[\|F(\hat{x}_k)\|^2\right] - \omega\gamma\mathbb{E}\left[\|F_{v_{k-1}}(\hat{x}_{k-1})\|^2\right] \\
&\quad - \omega(\gamma - \omega)\mathbb{E}\left[\|F_{v_k}(\hat{x}_k)\|^2\right] + \omega\gamma\mathbb{E}\left[\|F_{v_k}(\hat{x}_k) - F_{v_{k-1}}(\hat{x}_{k-1})\|^2\right] \\
&\leq \mathbb{E}\left[\|x_k - x^*\|^2\right] - \omega\gamma\mathbb{E}\left[\|F_{v_{k-1}}(\hat{x}_{k-1})\|^2\right] \\
&\quad - \omega(\gamma - 2\rho - \omega)\mathbb{E}\left[\|F_{v_k}(\hat{x}_k)\|^2\right] \\
&\quad + \omega\gamma\mathbb{E}\left[\|F_{v_k}(\hat{x}_k) - F_{v_{k-1}}(\hat{x}_{k-1})\|^2\right] \\
&\leq \mathbb{E}\left[\|x_k - x^*\|^2\right] - \omega\gamma\mathbb{E}\left[\|F_{v_{k-1}}(\hat{x}_{k-1})\|^2\right] \\
&\quad + \omega\gamma\mathbb{E}\left[\|F_{v_k}(\hat{x}_k) - F_{v_{k-1}}(\hat{x}_{k-1})\|^2\right], \tag{39}
\end{aligned}
$$

where we apply Jensen's inequality $\|F(\hat{x}_k)\|^2 = \|\mathbb{E}_{v_k}[F_{v_k}(\hat{x}_k)]\|^2 \leq \mathbb{E}_{v_k}[\|F_{v_k}(\hat{x}_k)\|^2]$ and $\gamma > 2\rho + \omega$. For $k = 0$ we have $x_1 = x_0 - \omega F_{v_0}(\hat{x}_0) = x_0 - \omega F_{v_0}(x_0)$ and

$$
\begin{aligned}
\mathbb{E}\left[\|x_1 - x^*\|^2\right] &= \|x_0 - x^*\|^2 - 2\omega\mathbb{E}\left[\langle x_0 - x^*, F_{v_0}(x_0)\rangle\right] + \omega^2\mathbb{E}\left[\|F_{v_0}(x_0)\|^2\right] \\
&= \|x_0 - x^*\|^2 - 2\omega\langle x_0 - x^*, F(x_0)\rangle + \omega^2\mathbb{E}\left[\|F_{v_0}(x_0)\|^2\right].
\end{aligned}
$$

Applying Weak Minty condition, we get

$$
\begin{aligned}
\mathbb{E}\left[\|x_1 - x^*\|^2\right] &= \|x_0 - x^*\|^2 + 2\omega\rho\|F(x_0)\|^2 + \omega^2\mathbb{E}\left[\|F_{v_0}(x_0)\|^2\right] \\
&\leq \|x_0 - x^*\|^2 + \omega(\omega + 2\rho)\mathbb{E}\left[\|F_{v_0}(x_0)\|^2\right]. \tag{40}
\end{aligned}
$$

The next step of our proof is in estimating the last term from (39). Using Young's inequality $\|a + b\|^2 \leq (1 + \alpha)\|a\|^2 + (1 + \alpha^{-1})\|b\|^2$, which holds for any $a, b \in \mathbb{R}^d$, $\alpha > 0$, we get for all $k \geq 2$

$$
\begin{aligned}
\mathbb{E}\left[\|F_{v_k}(\hat{x}_k) - F_{v_{k-1}}(\hat{x}_{k-1})\|^2\right] &\leq (1 + \alpha)\mathbb{E}\left[\|F(\hat{x}_k) - F(\hat{x}_{k-1})\|^2\right] \\
&\quad + (1 + \alpha^{-1})\mathbb{E}\left[\|F_{v_k}(\hat{x}_k) - F(\hat{x}_k)\right. \\
&\qquad \left. - (F_{v_{k-1}}(\hat{x}_{k-1}) - F(\hat{x}_{k-1}))\|^2\right] \\
&\leq (1 + \alpha)L^2\mathbb{E}\left[\|\hat{x}_k - \hat{x}_{k-1}\|^2\right] \\
&\quad + 2(1 + \alpha^{-1})\mathbb{E}\left[\|F_{v_k}(\hat{x}_k) - F(\hat{x}_k)\|^2\right. \\
&\qquad \left. + \|F_{v_{k-1}}(\hat{x}_{k-1}) - F(\hat{x}_{k-1})\|^2\right] \\
&\overset{(8)}{\leq} (1 + \alpha)L^2\mathbb{E}\left[\|\hat{x}_k - x_k + x_k - x_{k-1} + x_{k-1} - \hat{x}_{k-1}\|^2\right] \\
&\quad + 2(1 + \alpha^{-1})\delta\mathbb{E}\left[\|\hat{x}_k - x^*\|^2 + \|\hat{x}_{k-1} - x^*\|^2\right] \\
&\quad + 8(1 + \alpha^{-1})\sigma_*^2 \\
&\leq (1 + \alpha)L^2\mathbb{E}\left[\|(\gamma + \omega)F_{v_{k-1}}(\hat{x}_{k-1}) - \gamma F_{v_{k-2}}(\hat{x}_{k-2})\|^2\right] \\
&\quad + 4(1 + \alpha^{-1})\delta\mathbb{E}\left[\|x_k - x^*\|^2 + \|x_{k-1} - x^*\|^2\right] \\
&\quad + 4(1 + \alpha^{-1})\delta\gamma^2\mathbb{E}\left[\|F_{v_{k-1}}(\hat{x}_{k-1})\|^2 + \|F_{v_{k-2}}(\hat{x}_{k-2})\|^2\right] \\
&\quad + 8(1 + \alpha^{-1})\sigma_*^2 \\
&= (1 + \alpha)L^2(\gamma + \omega)^2\mathbb{E}\left[\|F_{v_{k-1}}(\hat{x}_{k-1})\|^2\right] \\
&\quad + (1 + \alpha)L^2\gamma^2\mathbb{E}\left[\|F_{v_{k-2}}(\hat{x}_{k-2})\|^2\right] \\
&\quad - 2(1 + \alpha)L^2\gamma(\gamma + \omega)\mathbb{E}\left[\langle F_{v_{k-1}}(\hat{x}_{k-1}), F_{v_{k-2}}(\hat{x}_{k-2})\rangle\right] \\
&\quad + 4(1 + \alpha^{-1})\delta\mathbb{E}\left[\|x_k - x^*\|^2 + \|x_{k-1} - x^*\|^2\right] \\
&\quad + 4(1 + \alpha^{-1})\delta\gamma^2\mathbb{E}\left[\|F_{v_{k-1}}(\hat{x}_{k-1})\|^2 + \|F_{v_{k-2}}(\hat{x}_{k-2})\|^2\right] \\
&\quad + 8(1 + \alpha^{-1})\sigma_*^2 \\
&= (1 + \alpha)L^2(\gamma + \omega)^2\mathbb{E}\left[\|F_{v_{k-1}}(\hat{x}_{k-1})\|^2\right] \\
&\quad + (1 + \alpha)L^2\gamma^2\mathbb{E}\left[\|F_{v_{k-2}}(\hat{x}_{k-2})\|^2\right] \\
&\quad - (1 + \alpha)L^2\gamma(\gamma + \omega)\mathbb{E}\left[\|F_{v_{k-1}}(\hat{x}_{k-1})\|^2 + \|F_{v_{k-2}}(\hat{x}_{k-2})\|^2\right] \\
&\quad + (1 + \alpha)L^2\gamma(\gamma + \omega)\mathbb{E}\left[\|F_{v_{k-1}}(\hat{x}_{k-1}) - F_{v_{k-2}}(\hat{x}_{k-2})\|^2\right] \\
&\quad + 4(1 + \alpha^{-1})\delta\mathbb{E}\left[\|x_k - x^*\|^2 + \|x_{k-1} - x^*\|^2\right] \\
&\quad + 4(1 + \alpha^{-1})\delta\gamma^2\mathbb{E}\left[\|F_{v_{k-1}}(\hat{x}_{k-1})\|^2 + \|F_{v_{k-2}}(\hat{x}_{k-2})\|^2\right] \\
&\quad + 8(1 + \alpha^{-1})\sigma_*^2 \\
&= (1 + \alpha)L^2\omega(\gamma + \omega)\mathbb{E}\left[\|F_{v_{k-1}}(\hat{x}_{k-1})\|^2\right] \\
&\quad - (1 + \alpha)L^2\gamma\omega\mathbb{E}\left[\|F_{v_{k-2}}(\hat{x}_{k-2})\|^2\right] \\
&\quad + (1 + \alpha)L^2\gamma(\gamma + \omega)\mathbb{E}\left[\|F_{v_{k-1}}(\hat{x}_{k-1}) - F_{v_{k-2}}(\hat{x}_{k-2})\|^2\right] \\
&\quad + 4(1 + \alpha^{-1})\delta\mathbb{E}\left[\|x_k - x^*\|^2 + \|x_{k-1} - x^*\|^2\right] \\
&\quad + 4(1 + \alpha^{-1})\delta\gamma^2\mathbb{E}\left[\|F_{v_{k-1}}(\hat{x}_{k-1})\|^2 + \|F_{v_{k-2}}(\hat{x}_{k-2})\|^2\right] \\
&\quad + 8(1 + \alpha^{-1})\sigma_*^2.
\end{aligned}
$$

Since $\hat{x}_0 = x_0$ and $\hat{x}_1 = x_1 - \gamma F_{v_0}(x_0) = x_0 - (\gamma + \omega)F_{v_0}(x_0)$, for $k = 1$ we have

$$
\begin{aligned}
\mathbb{E}\left[\|F_{v_1}(\hat{x}_1) - F_{v_0}(\hat{x}_0)\|^2\right] &= \mathbb{E}\left[\|F_{v_1}(\hat{x}_1) - F_{v_0}(x_0)\|^2\right] \\
&\leq (1 + \alpha)\mathbb{E}\left[\|F(\hat{x}_1) - F(x_0)\|^2\right] \\
&\quad + (1 + \alpha^{-1})\mathbb{E}\left[\|F_{v_1}(\hat{x}_1) - F(\hat{x}_1) - (F_{v_0}(x_0) - F(x_0))\|^2\right] \\
&\leq (1 + \alpha)L^2\mathbb{E}\left[\|\hat{x}_1 - x_0\|^2\right] \\
&\quad + 2(1 + \alpha^{-1})\mathbb{E}\left[\|F_{v_1}(\hat{x}_1) - F(\hat{x}_1)\|^2 + \|F_{v_0}(x_0) - F(x_0)\|^2\right]
\end{aligned}
$$

Then using (8) we get,

$$\mathbb{E}\left[\|F_{v_1}(\hat{x}_1) - F_{v_0}(\hat{x}_0)\|^2\right] \overset{(8)}{\leq} (1+\alpha)L^2(\gamma+\omega)^2\mathbb{E}\left[\|F_{v_0}(x_0)\|^2\right]$$
$$+2(1+\alpha^{-1})\delta\mathbb{E}\left[\|\hat{x}_1 - x^*\|^2 + \|x_0 - x^*\|^2\right] + 8(1+\alpha)\sigma_*^2$$
$$\leq \left((1+\alpha)L^2 + 4(1+\alpha^{-1})\delta\right)(\gamma+\omega)^2\mathbb{E}\left[\|F_{v_0}(x_0)\|^2\right]$$
$$+6(1+\alpha^{-1})\delta\|x_0 - x^*\|^2 + 8(1+\alpha)\sigma_*^2.$$

Let $\{w_k\}_{k=0}^{K-1}$ be a non-increasing sequence of positive numbers that will be specified later and $W_K = \sum_{k=0}^{K-1} w_k$. Summing up the above two inequalities with weights $\{w_k\}_{k=1}^{K-1}$, we derive

$$\sum_{k=1}^{K-1} w_k\mathbb{E}\left[\|F_{v_k}(\hat{x}_k) - F_{v_{k-1}}(\hat{x}_{k-1})\|^2\right] \leq (1+\alpha)L^2 \sum_{k=1}^{K-3}\left(\omega(\gamma+\omega)w_{k+1}\mathbb{E}\left[\|F_{v_k}(\hat{x}_k)\|^2\right]\right.$$
$$-\gamma\omega w_{k+2}\mathbb{E}\left[\|F_{v_k}(\hat{x}_k)\|^2\right]\Big)$$
$$+(1+\alpha)L^2\omega(\gamma+\omega)w_{K-1}\mathbb{E}\left[\|F_{v_{K-2}}(\hat{x}_{K-2})\|^2\right]$$
$$-(1+\alpha)L^2\gamma\omega w_2\mathbb{E}\left[\|F_{v_0}(x_0)\|^2\right]$$
$$+(1+\alpha)L^2\gamma(\gamma+\omega)\sum_{k=1}^{K-2} w_{k+1}\mathbb{E}\left[\|F_{v_k}(\hat{x}_k)\right.$$
$$-F_{v_{k-1}}(\hat{x}_{k-1})\|^2\Big] + 4(1+\alpha^{-1})\delta\sum_{k=2}^{K-1} w_k\mathbb{E}\left[\|x_k - x^*\|^2\right]$$
$$+w_k\mathbb{E}\left[\|x_{k-1} - x^*\|^2\right]$$
$$+4(1+\alpha^{-1})\delta\gamma^2\sum_{k=1}^{K-2} w_{k+1}\mathbb{E}\left[\|F_{v_k}(\hat{x}_k)\|^2\right.$$
$$+\|F_{v_{k-1}}(\hat{x}_{k-1})\|^2\Big] + 8(1+\alpha^{-1})(W_K - w_0 - w_1)\sigma_*^2$$
$$+\left((1+\alpha)L^2 + 4(1+\alpha^{-1})\delta\right)(\gamma+\omega)^2 w_1\mathbb{E}\left[\|F_{v_0}(x_0)\|^2\right]$$
$$+6(1+\alpha^{-1})\delta w_1\|x_0 - x^*\|^2 + 8(1+\alpha)w_1\sigma_*^2.$$

Next, we rearrange the terms using $w_k \geq w_{k+1}$ and new notation $\Delta_k = \mathbb{E}\left[\|F_{v_k}(\hat{x}_k) - F_{v_{k-1}}(\hat{x}_{k-1})\|^2\right]$:

$$\left(1 - (1+\alpha)L^2\gamma(\gamma+\omega)\right)\sum_{k=1}^{K-1} w_k\Delta_k \leq \sum_{k=1}^{K-2}(1+\alpha)L^2\omega(\gamma+\omega)w_k\mathbb{E}\left[\|F_{v_k}(\hat{x}_k)\|^2\right]$$
$$+8(1+\alpha^{-1})\delta\gamma^2 w_k\mathbb{E}\left[\|F_{v_k}(\hat{x}_k)\|^2\right]$$
$$+\left((1+\alpha)L^2 + 8(1+\alpha^{-1})\delta\right)(\gamma+\omega)^2 w_0\mathbb{E}\left[\|F_{v_0}(x_0)\|^2\right]$$
$$+12(1+\alpha^{-1})\delta\sum_{k=1}^{K-1} w_k\mathbb{E}\left[\|x_k - x^*\|^2\right]$$
$$+8(1+\alpha^{-1})(W_K - w_0)\sigma_*^2.$$

To simplify the above inequality we choose $\alpha = \frac{1}{2L^2\gamma(\gamma+\omega)} - \frac{1}{2}$, which is positive due to $\gamma < 1/L$ and $\gamma + \omega < 1/L$. In this case, we have

$$(1+\alpha)L^2\gamma(\gamma+\omega) = \frac{1}{2}L^2\gamma(\gamma+\omega) + \frac{1}{2},$$
$$(1+\alpha)L^2(\gamma+\omega)^2 = \frac{1}{2}L^2(\gamma+\omega)^2 + \frac{\gamma+\omega}{2\gamma} \leq \frac{3}{2},$$
$$(1+\alpha)L^2\omega(\gamma+\omega) = \frac{1}{2}L^2\omega(\gamma+\omega) + \frac{\omega}{2\gamma} = \frac{L\omega}{2}\left(L(\gamma+\omega) + \frac{1}{\gamma L}\right) \leq \frac{3L\omega}{2},$$
$$1+\alpha^{-1} = 1 + \frac{2L^2\gamma(\gamma+\omega)}{1 - L^2\gamma(\gamma+\omega)} = \frac{1 + L^2\gamma(\gamma+\omega)}{1 - L^2\gamma(\gamma+\omega)} \leq \frac{2}{1 - L^2\gamma(\gamma+\omega)},$$

where we also use $1/2L < \gamma < 1/L$ and $\gamma + \omega < 1/L$. Using these relations, we can continue our derivation as follows:

$$\frac{1}{2}\left(1 - L^2\gamma(\gamma+\omega)\right)\sum_{k=1}^{K-1} w_k\Delta_k \leq \sum_{k=1}^{K-2}\left(\frac{3L\omega}{2} + \frac{16}{1-L^2\gamma(\gamma+\omega)}\delta\gamma^2\right)w_k\mathbb{E}\left[\|F_{v_k}(\hat{x}_k)\|^2\right]$$

$$+\left(\frac{3}{2} + \frac{16}{1-L^2\gamma(\gamma+\omega)}\delta(\gamma+\omega)^2\right)w_0\mathbb{E}\left[\|F_{v_0}(x_0)\|^2\right]$$

$$+\frac{24}{1-L^2\gamma(\gamma+\omega)}\delta\sum_{k=1}^{K-1} w_k\mathbb{E}\left[\|x_k - x^*\|^2\right]$$

$$+\frac{16}{1-L^2\gamma(\gamma+\omega)}(W_K - w_0)\sigma_*^2.$$

Dividing both sides by $\frac{1}{2}\left(1 - L^2\gamma(\gamma+\omega)\right)$, we derive

$$\sum_{k=1}^{K-1} w_k\Delta_k \leq \sum_{k=1}^{K-2}\left(\frac{3L\omega}{1-L^2\gamma(\gamma+\omega)} + \frac{32}{(1-L^2\gamma(\gamma+\omega))^2}\delta\gamma^2\right)w_k\mathbb{E}\left[\|F_{v_k}(\hat{x}_k)\|^2\right]$$

$$+\left(\frac{3}{1-L^2\gamma(\gamma+\omega)} + \frac{32}{(1-L^2\gamma(\gamma+\omega))^2}\delta(\gamma+\omega)^2\right)w_0\mathbb{E}\left[\|F_{v_0}(x_0)\|^2\right]$$

$$+\frac{48}{(1-L^2\gamma(\gamma+\omega))^2}\delta\sum_{k=1}^{K-1} w_k\mathbb{E}\left[\|x_k - x^*\|^2\right]$$

$$+\frac{32}{(1-L^2\gamma(\gamma+\omega))^2}(W_K - w_0)\sigma_*^2$$

$$= \sum_{k=1}^{K-2} C_1 w_k\mathbb{E}\left[\|F_{v_k}(\hat{x}_k)\|^2\right] + C_2 w_0\mathbb{E}\left[\|F_{v_0}(x_0)\|^2\right]$$

$$+3C_3\delta\sum_{k=1}^{K-1} w_k\mathbb{E}\left[\|x_k - x^*\|^2\right] + 2C_3 W_K\sigma_*^2, \tag{41}$$

where $C_1 = \frac{3L\omega}{1-L^2\gamma(\gamma+\omega)} + \frac{32}{(1-L^2\gamma(\gamma+\omega))^2}\delta\gamma^2$, $C_2 = \frac{3}{1-L^2\gamma(\gamma+\omega)} + \frac{32}{(1-L^2\gamma(\gamma+\omega))^2}\delta(\gamma+\omega)^2$, and $C_3 = \frac{16}{(1-L^2\gamma(\gamma+\omega))^2}$. Summing up inequalities (39) for $k = 1, \ldots, K-1$ with weights $w_1, \ldots, w_{K-1}$ and (40) with weight $w_0$, we get

$$\sum_{k=0}^{K-1} w_k\mathbb{E}\left[\|x_{k+1} - x^*\|^2\right] \leq \sum_{k=0}^{K-1} w_k\mathbb{E}\left[\|x_k - x^*\|^2\right] - \omega\gamma\sum_{k=1}^{K-1} w_k\mathbb{E}\left[\|F_{v_{k-1}}(\hat{x}_{k-1})\|^2\right]$$

$$+\omega\gamma\sum_{k=1}^{K-1} w_k\Delta_k + \omega(\omega+2\rho)w_0\mathbb{E}\left[\|F_{v_0}(x_0)\|^2\right].$$

Since $w_k \geq w_{k+1}$, we can continue the derivation as follows:

$$\sum_{k=0}^{K-1} w_k\mathbb{E}\left[\|x_{k+1} - x^*\|^2\right] \leq \sum_{k=0}^{K-1} w_k\mathbb{E}\left[\|x_k - x^*\|^2\right] - \omega\gamma\sum_{k=0}^{K-2} w_k\mathbb{E}\left[\|F_{v_k}(\hat{x}_k)\|^2\right]$$

$$+\omega\gamma\sum_{k=1}^{K-1} w_k\Delta_k + \omega(\omega+2\rho)w_0\mathbb{E}\left[\|F_{v_0}(x_0)\|^2\right]$$

$$\overset{(41)}{\leq} \sum_{k=0}^{K-1}(1 + 3C_3\omega\gamma\delta)w_k\mathbb{E}\left[\|x_k - x^*\|^2\right]$$

$$-\omega\gamma(1 - C_1)\sum_{k=0}^{K-2} w_k\mathbb{E}\left[\|F_{v_k}(\hat{x}_k)\|^2\right]$$

$$+2\omega\gamma C_2 w_0\mathbb{E}\left[\|F_{v_0}(\hat{x}_0)\|^2\right] + 2\omega\gamma C_3 W_K\sigma_*^2.$$

Now we need to specify the weights $w_{-1}, w_0, w_1, \ldots, w_{K-1}$. Let $w_{K-2} = 1$ and $w_{k-1} = (1 + 3C_3\omega\gamma\delta)w_k$. Then, rearranging the terms, dividing both sides by $\omega\gamma(1 - C_1)W_{K-1}$, we get

$$
\begin{aligned}
\min_{0 \leq k \leq K-1} \mathbb{E}\left[\|F(\hat{x}_k)\|^2\right] &\leq \min_{0 \leq k \leq K-1} \mathbb{E}\left[\|F_{v_k}(\hat{x}_k)\|^2\right] \\
&\leq \sum_{k=0}^{K-2} \frac{w_k}{W_{K-1}} \mathbb{E}\left[\|F_{v_k}(\hat{x}_k)\|^2\right] \\
&\leq \frac{1}{\omega\gamma(1 - C_1)W_{K-1}} \sum_{k=0}^{K-1} \left(w_{k-1}\mathbb{E}\left[\|x_k - x^*\|^2\right]\right. \\
&\quad \left. -w_k\mathbb{E}\left[\|x_{k+1} - x^*\|^2\right]\right) + \frac{2C_2 w_0 \mathbb{E}\left[\|F_{v_0}(\hat{x}_0)\|^2\right]}{(1 - C_1)W_{K-1}} \\
&\quad + \frac{2C_3 W_K \sigma_*^2}{(1 - C_1)W_{K-1}} \\
&\leq \frac{w_{-1}\|x_0 - x^*\|^2}{\omega\gamma(1 - C_1)W_{K-1}} + \frac{2C_2 w_0 \mathbb{E}\left[\|F_{v_0}(\hat{x}_0)\|^2\right]}{(1 - C_1)W_{K-1}} + \frac{2C_3 W_K \sigma_*^2}{(1 - C_1)W_{K-1}}.
\end{aligned}
$$

It remains to simplify the right-hand side of the above inequality. First, we notice that $W_{K-1} = \sum_{k=0}^{K-2} w_k \geq (K-1)w_{K-2} = K - 1$ since $w_k \geq w_{k+1}$. Moreover, $w_{-1} = (1 + 3C_3\omega\gamma\delta)^{K-1}$. Next,

$$
\begin{aligned}
C_1 &= \frac{3L\omega}{1 - L^2\gamma(\gamma + \omega)} + \frac{32}{(1 - L^2\gamma(\gamma + \omega))^2}\delta\gamma^2 \\
&\leq \frac{3L\omega}{1 - L\gamma} + \frac{32}{(1 - L\gamma)^2} \cdot \frac{(1 - L\gamma)L^3\omega}{32} \cdot \gamma^2 \leq \frac{4L\omega}{1 - L\gamma}, \\
C_2 &= \frac{3}{1 - L^2\gamma(\gamma + \omega)} + \frac{32}{(1 - L^2\gamma(\gamma + \omega))^2}\delta(\gamma + \omega)^2 \\
&\leq \frac{3}{1 - L\gamma} + \frac{32}{(1 - L\gamma)^2} \cdot \frac{(1 - L\gamma)L^3\omega}{32} \cdot (\gamma + \omega)^2 \leq \frac{4}{1 - L\gamma}, \\
C_3 &= \frac{16}{(1 - L^2\gamma(\gamma + \omega))^2} \leq \frac{16}{(1 - L\gamma)^2},
\end{aligned}
$$

where we use $\delta \leq {(1-L\gamma)L^3\omega}/{16}$ and $\gamma + \omega < {1}/{L}$. Using these inequalities, we simplify the bound as follows:

$$
\begin{aligned}
\min_{0 \leq k \leq K-1} \mathbb{E}\left[\|F(\hat{x}_k)\|^2\right] &\leq \frac{(1 - L\gamma)(1 + 3C_3\omega\gamma\delta)^{K-1}\|x_0 - x^*\|^2}{\omega\gamma(1 - L(\gamma + 4\omega))(K - 1)} \\
&\quad + \frac{8(1 + 3C_3\omega\gamma\delta)^{K-2}\mathbb{E}\left[\|F_{v_0}(\hat{x}_0)\|^2\right]}{(1 - L(\gamma + 4\omega))(K - 1)} \\
&\quad + \frac{32\sigma_*^2}{(1 - L\gamma)(1 - L(\gamma + 4\omega))} \\
&\leq \frac{(1 - L\gamma)\left(1 + \frac{48\omega\gamma\delta}{(1-L\gamma)^2}\right)^{K-1}\|x_0 - x^*\|^2}{\omega\gamma(1 - L(\gamma + 4\omega))(K - 1)} \\
&\quad + \frac{8\left(1 + \frac{48\omega\gamma\delta}{(1-L\gamma)^2}\right)^{K-2}\mathbb{E}\left[\|F_{v_0}(\hat{x}_0)\|^2\right]}{(1 - L(\gamma + 4\omega))(K - 1)} \\
&\quad + \frac{32\sigma_*^2}{(1 - L\gamma)(1 - L(\gamma + 4\omega))} \quad (42)
\end{aligned}
$$

where we use $W_K = W_{K-1} + w_{K-1} \leq W_{K-1} + w_{K-2} \leq 2W_{K-1}$. Finally, we use (8) to upper-bound $\mathbb{E}\left[\|F_{v_0}(\hat{x}_0)\|^2\right]$:

$$
\begin{aligned}
\mathbb{E}\left[\|F_{v_0}(\hat{x}_0)\|^2\right] &= \mathbb{E}\left[\|F_{v_0}(x_0)\|^2\right] \overset{(8)}{\leq} \delta\|x_0 - x^*\|^2 + \|F(x_0)\|^2 + 2\sigma_*^2 \\
&\leq (\delta + L^2)\|x_0 - x^*\|^2 + 2\sigma_*^2.
\end{aligned}
$$

Plugging this inequality in (42), we derive

$$
\min_{0 \leq k \leq K-1} \mathbb{E}\left[\|F(\hat{x}_k)\|^2\right] \leq \frac{\left(1 + 8\omega\gamma(\delta + L^2) - L\gamma\right)\left(1 + \frac{48\omega\gamma\delta}{(1-L\gamma)^2}\right)^{K-1}\|x_0 - x^*\|^2}{\omega\gamma(1 - L(\gamma + 4\omega))(K-1)}
$$
$$
+ \frac{4\left(8 + \frac{1-L\gamma}{K-1}\left(1 + \frac{48\omega\gamma\delta}{(1-L\gamma)^2}\right)^{K-1}\right)\sigma_*^2}{(1 - L\gamma)(1 - L(\gamma + 4\omega))},
$$

which concludes the proof. $\qquad\square$

## E.5   Proof of Theorem 4.5

**Theorem E.5.** Let $F$ be $L$-Lipschitz and satisfy Weak Minty condition with parameter $\rho < 1/(2L)$. Assume that inequality (8) holds (e.g., it holds whenever Assumption 3.1 holds, see Lemma 3.2). Assume that $\gamma_k = \gamma$, $\omega_k = \omega$ and

$$
\max\left\{2\rho, \frac{1}{2L}\right\} < \gamma < \frac{1}{L}, \quad 0 < \omega < \min\left\{\gamma - 2\rho, \frac{1}{4L} - \frac{\gamma}{4}\right\}.
$$

Then, for all $K \geq 2$ the iterates produced by mini-batched SPEG with batch-size

$$
\tau \geq \max\left\{1, \frac{32\delta}{(1-L\gamma)L^3\omega}, \frac{48\omega\gamma\delta(K-1)}{(1-L\gamma)^2}, \frac{2\omega\gamma\sigma_*^2(K-1)}{(1-L\gamma)\|x_0 - x^*\|^2}\right\} \tag{43}
$$

satisfy

$$
\min_{0 \leq k \leq K-1} \mathbb{E}\left[\|F(\hat{x}_k)\|^2\right] \leq \frac{48\|x_0 - x^*\|^2}{\omega\gamma(1 - L(\gamma + 4\omega))(K-1)}. \tag{44}
$$

*Proof.* Mini-batched SPEG uses estimator

$$
F_{v_k}(\hat{x}_k) = \frac{1}{\tau}\sum_{i=1}^{\tau} F_{v_{k,i}}(\hat{x}_k),
$$

where $F_{v_{k,1}}(\hat{x}_k), \ldots, F_{v_{k,\tau}}(\hat{x}_k)$ are independent samples satisfying (8) with parameters $\delta$ and $\sigma_*^2$. Using variance decomposition and independence of $F_{v_{k,1}}(\hat{x}_k), \ldots, F_{v_{k,\tau}}(\hat{x}_k)$, we get

$$
\begin{aligned}
\mathbb{E}_{v_k}\left[\|F_{v_k}(\hat{x}_k)\|^2\right] &= \mathbb{E}_{v_k}\left[\|F_{v_k}(\hat{x}_k) - F(\hat{x}_k)\|^2\right] + \|F(\hat{x}_k)\|^2 \\
&= \mathbb{E}_{v_k}\left[\left\|\frac{1}{\tau}\sum_{i=1}^{b}(F_{v_{k,i}}(\hat{x}_k) - F(\hat{x}_k))\right\|^2\right] + \|F(\hat{x}_k)\|^2 \\
&= \frac{1}{\tau^2}\sum_{i=1}^{\tau}\mathbb{E}_{v_k}\left[\|F_{v_{k,i}}(\hat{x}_k) - F(\hat{x}_k)\|^2\right] + \|F(\hat{x}_k)\|^2 \\
&\overset{(8)}{\leq} \frac{\delta}{\tau}\|\hat{x}_k - x^*\|^2 + \|F(\hat{x}_k)\|^2 + \frac{2\sigma_*^2}{\tau}.
\end{aligned}
$$

That is, mini-batched estimator $F_{v_k}(\hat{x}_k)$ satisfies (8) with parameters $\delta/\tau$ and $\sigma_*^2/\tau$. Therefore, Theorem E.4 implies

$$
\min_{0 \leq k \leq K-1} \mathbb{E}\left[\|F(\hat{x}_k)\|^2\right] \leq \frac{\left(1 + 4\omega\gamma\left(\frac{\delta}{\tau} + L^2\right) - L\gamma\right)\left(1 + \frac{48\omega\gamma\delta}{(1-L\gamma)^2\tau}\right)^{K-1}\|x_0 - x^*\|^2}{\omega\gamma(1 - L(\gamma + 4\omega))(K-1)}
$$
$$
+ \frac{8\left(8 + \frac{1-L\gamma}{K-1}\left(1 + \frac{48\omega\gamma\delta}{(1-L\gamma)^2\tau}\right)^{K-1}\right)\sigma_*^2}{(1 - L\gamma)(1 - L(\gamma + 4\omega))\tau}. \tag{45}
$$

Since $\tau$ satisfies (43) and $\gamma \leq 1/L$, $\omega \leq 1/4L$, we have

$$4\omega\gamma\left(\frac{\delta}{\tau} + L^2\right) \leq \frac{1}{4L^2}\left(\delta \cdot \frac{(1-L\gamma)L^3\omega}{16\delta} + L^2\right) \leq 1,$$

$$\left(1 + \frac{48\omega\gamma\delta}{(1-L\gamma)^2\tau}\right)^{K-1} \leq \left(1 + \frac{48\omega\gamma\delta}{(1-L\gamma)^2} \cdot \frac{(1-L\gamma)^2}{48\omega\gamma\delta(K-1)}\right)^{K-1}$$

$$= \left(1 + \frac{1}{K-1}\right)^{K-1} \leq \exp(1) < 3.$$

Using this, we can simplify (45) as follows:

$$\min_{0 \leq k \leq K-1} \mathbb{E}\left[\|F(\hat{x}_k)\|^2\right] \leq \frac{6\|x_0 - x^*\|^2}{\omega\gamma(1 - L(\gamma + 4\omega))(K-1)} + \frac{88\sigma_*^2}{(1-L\gamma)(1 - L(\gamma + 4\omega))\tau}$$

$$\overset{(43)}{\leq} \frac{6\|x_0 - x^*\|^2}{\omega\gamma(1 - L(\gamma + 4\omega))(K-1)}$$

$$+ \frac{88\sigma_*^2}{(1-L\gamma)(1 - L(\gamma + 4\omega))} \cdot \frac{(1-L\gamma)\|x_0 - x^*\|^2}{2\omega\gamma\sigma_*^2}$$

$$= \frac{48\|x_0 - x^*\|^2}{\omega\gamma(1 - L(\gamma + 4\omega))(K-1)}.$$

This concludes the proof. $\qquad\square$

**On Oracle Complexity of Theorem 4.5.** Let us now express the result of Theorem 4.5 via oracle complexity.

Oracle complexity captures the computational requirements required to solve a specific optimization problem. That is, given a prespecified accuracy $\varepsilon > 0$, it measures the number of oracle calls needed to solve the problem to this $\varepsilon$ accuracy. In our setting, an oracle call indicates the computation of one operator, $F_i$ (for some $i \in [n]$). Therefore, in Theorem 4.5, where a mini-batch of size $\tau$ is required in each iteration of the update rule, we have $\tau$ many oracle calls per iteration. In that scenario, the total number of oracle calls required to obtain specific accuracy $\varepsilon > 0$ is given by $K\tau$ (multiplication of $K$ iterations with $\tau$ oracle calls).

Note that according to Theorem 4.5 to achieve an $\varepsilon$ accuracy, we need $K \geq \frac{C\|x_0 - x^*\|^2}{\epsilon}$ iterations. This follows trivially by

$$\min_{0 \leq k \leq K-1} \mathbb{E}\left[\|F(\hat{x}_k)\|^2\right] \overset{\text{Theorem 4.5}}{\leq} \frac{C\|x_0 - x^*\|^2}{K-1} \leq \varepsilon. \tag{46}$$

Therefore, using $K \geq \frac{C\|x_0 - x^*\|^2}{\epsilon}$ in combination with the lower bound on $\tau$ from (15), the total number of oracle calls to satisfy (46) is given by:

$$K\tau \geq \max\left\{\frac{C\|x_0 - x^*\|^2}{\epsilon}, \frac{32C\delta\|x_0 - x^*\|^2}{(1-L\gamma)L^3\omega\epsilon}, \frac{48C^2\omega\gamma\delta\|x_0 - x^*\|^4}{(1-\gamma L)^2\epsilon^2}, \frac{2C^2\omega\gamma\sigma_*^2\|x_0 - x^*\|^2}{(1-L\gamma)\epsilon^2}\right\}.$$

## F  Further Results on Arbitrary Sampling

### F.1  Proof of Proposition 5.1

Expanding the left hand side of Expected Residual (ER) condition we have

$$\mathbb{E}\|(F_v(x) - F_v(x^*)) - (F(x) - F(x^*))\|^2 \overset{(25)}{=} \mathbb{E}\|(F_v(x) - F_v(x^*))\|^2 - \|F(x) - F(x^*)\|^2$$
$$\leq \mathbb{E}\|F_v(x) - F_v(x^*)\|^2. \qquad (47)$$

For any $x$ and $y$ with $v_i = \frac{1}{p_i}$ we obtain

$$\|F_v(x) - F_v(y)\|^2 = \frac{1}{n^2}\left\|\sum_{i \in S}\frac{1}{p_i}(F_i(x) - F_i(y))\right\|^2$$
$$= \sum_{i,j \in S}\left\langle\frac{1}{np_i}(F_i(x) - F_i(y)), \frac{1}{np_j}(F_j(x) - F_j(y))\right\rangle.$$

Then taking expectation on both sides we get

$$\mathbb{E}\|F_v(x) - F_v(y)\|^2 = \sum_C p_C \sum_{i,j \in C}\left\langle\frac{1}{np_i}(F_i(x) - F_i(y)), \frac{1}{np_j}(F_j(x) - F_j(y))\right\rangle$$
$$= \sum_{i,j=1}^n \sum_{C:i,j \in C} p_C\left\langle\frac{1}{np_i}(F_i(x) - F_i(y)), \frac{1}{np_j}(F_j(x) - F_j(y))\right\rangle$$
$$= \sum_{i,j=1}^n \frac{P_{ij}}{p_i p_j}\left\langle\frac{1}{n}(F_i(x) - F_i(y)), \frac{1}{n}(F_j(x) - F_j(y))\right\rangle.$$

Now we consider the case, where the ratio $\frac{P_{ij}}{p_i p_j} = c_2$ i.e. constant for $i \neq j$ and $P_{ii} = p_i$. Then from the above computations we derive

$$\mathbb{E}\|F_v(x) - F_v(y)\|^2 = \sum_{i \neq j}^n c_2\left\langle\frac{1}{n}(F_i(x) - F_i(y)), \frac{1}{n}(F_i(x) - F_i(y))\right\rangle$$
$$+ \sum_{i=1}^n \frac{1}{n^2 p_i}\|F_i(x) - F_i(y)\|^2$$
$$= \sum_{i,j=1}^n c_2\left\langle\frac{1}{n}(F_i(x) - F_i(y)), \frac{1}{n}(F_i(x) - F_i(y))\right\rangle$$
$$+ \sum_{i=1}^n \frac{1 - p_i c_2}{n^2 p_i}\|F_i(x) - F_i(y)\|^2$$
$$\overset{(19)}{\leq} c_2\|F(x) - F(y)\|^2 + \sum_{i=1}^n \frac{1 - p_i c_2}{n^2 p_i}L_i^2\|x - y\|^2$$
$$\overset{(18)}{\leq} \left(c_2 L^2 + \frac{1}{n^2}\sum_{i=1}^n \frac{1 - p_i c_2}{p_i}L_i^2\right)\|x - y\|^2.$$

Thus replacing $y = x^*$ and combining with (47) we get the following bound on the Expected Residual:

$$\mathbb{E}\|(F_v(x) - F_v(x^*)) - (F(x) - F(x^*))\|^2 \leq \left(c_2 L^2 + \frac{1}{n^2}\sum_{i=1}^n \frac{1 - p_i c_2}{p_i}L_i^2\right)\|x - x^*\|^2. \quad (48)$$

For single-element sampling $c_2 = 0$ (as probability of two points appearing in same sample is zero for single element sampling i.e. $P_{ij} = 0$). Then we obtain

$$\delta \leq \frac{2}{n^2}\sum_{i=1}^n \frac{L_i^2}{p_i}$$

from (48). This completes the derivation of $\delta$ for single element sampling. To compute $\sigma_*^2$ for single element sampling, we replace

$$P_{ij} = \begin{cases} p_i & \text{if } i = j \\ 0 & \text{otherwise} \end{cases}$$

in (30) to get

$$\sigma_*^2 = \frac{1}{n^2} \sum_{i=1}^{n} \frac{1}{p_i} \|F_i(x^*)\|^2.$$

# G   Numerical Experiments

In Appendix G.1, we add more details on the experiments discussed in the main paper. Furthermore, in Appendix G.2, we run more experiments to evaluate the performance of SPEG on quasi-strongly monotone and weak MVI problems.

## G.1   More Details on the Numerical Experiments of Section 6

**On Constant vs Switching Stepsize Rule.**   We run the experiments on two synthetic datasets. In Fig. 1 of the main paper, we take $\mu_A = \mu_C = 0.6$. Here we include one more plot with a similar flavor but in a different setting. For Fig. 5, we generate the data such that eigenvalues of $A_1, B_1, C_1$ are generated uniformly from the interval $[0.1, 10]$. In the new plot, similar to the main paper, we can see the benefit of switching the step-size rule of Theorem 4.3.

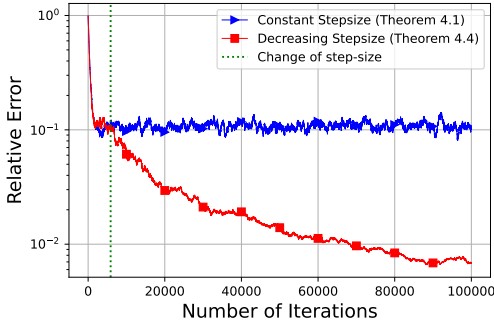

Figure 5: Comparison of the constant step-size rule (9) with the switching step-sizes (11) on the strongly monotone quadratic game.

**On Weak Minty VIPs.**   In this experiment, we generate $\xi_i, \zeta_i$ such that $\frac{1}{n}\sum_{i=1}^{n}\xi_i = \sqrt{63}$ and $\frac{1}{n}\sum_{i=1}^{n}\zeta_i = -1$. This choice of $\xi_i, \zeta_i$ ensures that $L = 8$ and $\rho = 1/32$ for the min-max problem we considered in Section 6.2. In Fig. 6, we again implement the SPEG on (17) with batchsize = $0.15 \times n$ (different batchsize compare to the plot of the main paper).

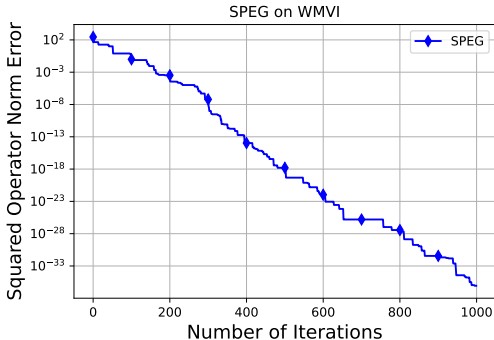

Figure 6: Trajectory of SPEG for solving weak MVI using a batchsize = $0.15 \times n$.

## G.2   Additional Experiments

In this subsection, we include more experiments to evaluate the performance of SPEG on quasi-strongly monotone and weak MVI problems. First, we run the experiment comparing constant and switching step-size rules on a different setup than the one we included in the main paper to analyze the performance of SPEG under different condition numbers. Then, we implement SPEG on the weak MVI of (17). To evaluate the performance in this experiment, we plot $\|F(\hat{x}_k)\|^2/\|F(x_0)\|^2$ on the $y$-axis.

### G.2.1 Strongly Monotone Quadratic Game:

In this experiment, we compare the proposed constant step-size (9) and the switching step-size rule (11). We implement our algorithm on operator $F : \mathbb{R}^4 \to \mathbb{R}^4$ given by

$$F(x) := \frac{1}{3} \left( M_1(x - x_1^*) + M_2(x - x_2^*) + M_3(x - x_3^*) \right),$$

where $M_1$, $M_2$ and $M_3$ are the diagonal matrices,

$$M_1 = \begin{pmatrix} \Delta & & & \\ & 1 & & \\ & & 1 & \\ & & & 1 \end{pmatrix}, \quad M_2 = \begin{pmatrix} 1 & & & \\ & \Delta & & \\ & & 1 & \\ & & & 1 \end{pmatrix}, \quad M_3 = \begin{pmatrix} 1 & & & \\ & 1 & & \\ & & \Delta & \\ & & & 1 \end{pmatrix}$$

and

$$x_1^* = \begin{pmatrix} \Delta \\ 0 \\ 0 \\ \Delta \end{pmatrix}, \quad x_2^* = \begin{pmatrix} 0 \\ \Delta \\ 0 \\ 0 \end{pmatrix}, \quad x_3^* = \begin{pmatrix} 0 \\ 0 \\ \Delta \\ 0 \end{pmatrix}.$$

This choice of $M_i$ and $x_i^*$ ensures that the Lipschitz constant of operator $F$ is $\frac{\Delta+2}{3}$ while quasi-strong monotonicity parameter (3) is $\mu = 1$. Hence the condition number of $F$ is given by $\frac{\Delta+2}{3}$. This allows us to vary the condition number of operator $F$ by changing the value of $\Delta$. For Fig. 7a we take $\Delta = 3$ (condition number = 1.67) while for Fig. 7b we choose $\Delta = 10$ (condition number = 10.67). The vertical dotted line in plots of Fig. 7 marks the transition point from constant to switching step-size rule as predicted by our theoretical result in Theorem 4.3.

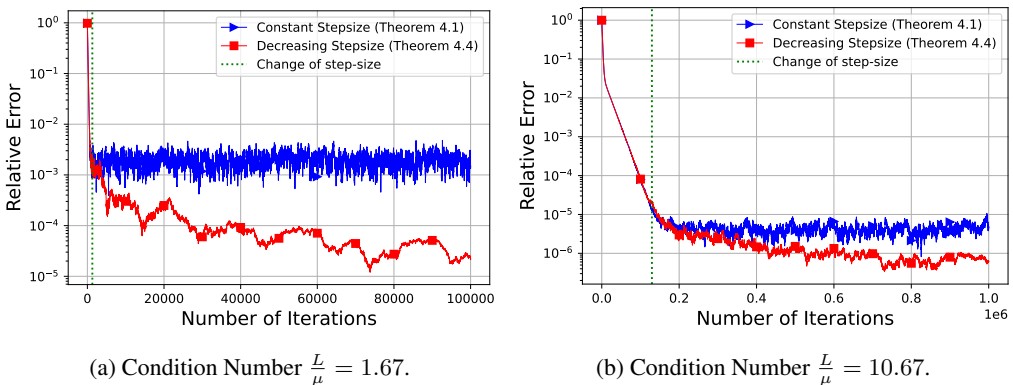

(a) Condition Number $\frac{L}{\mu} = 1.67$.        (b) Condition Number $\frac{L}{\mu} = 10.67$.

Figure 7: *Illustration of switching rule (11) in Theorem 4.3. The dotted line marks the transition from phase 1 (where we use constant step-size) to phase 2 (where we use decreasing step-size).*

### G.2.2 Weak Minty VIPs Continued

In this experiment, we reevaluate the performance of SPEG on weak MVI example of (17). That is, we generate the data in exactly the same way as the ones in section 6.2 with $n = 100$. In Fig. 8a and 8b, we implement SPEG with batchsize 10 and 15, respectively (we note that in this setting the full-gradient evaluation requires a batchsize of 100). For these plots, we use the relative operator norm on the y-axis, i.e. $\|F(\hat{x}_k)\|^2 / \|F(x_0)\|^2$, where $x_0$ denotes the starting point of SPEG. As expected, the plots illustrate that SPEG performs better as we increase the batchsize. From Fig. 8 it is clear that with batchsize 15 SPEG reaches an accuracy close to $10^{-10}$ while when we use a batchsize of 10 for the same number of iterations we are only able to converge to an accuracy of $10^{-4}$.

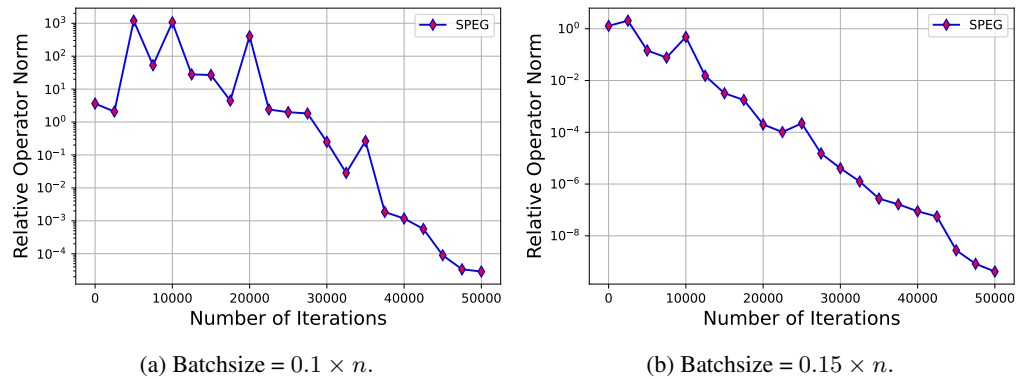

(a) Batchsize = $0.1 \times n$.

(b) Batchsize = $0.15 \times n$.

Figure 8: *Performance of* SPEG *for solving weak MVI with different batchsizes. In plot (a) we use a batchsize of* 10 *while in plot (b) we use* 15.

