# OpenReview forum: "Single-Call Stochastic Extragradient Methods for Structured Non-monotone Variational Inequalities: Improved Analysis under Weaker Conditions"
_NeurIPS.cc/2023/Conference — NeurIPS 2023 poster_

### Official Review · Reviewer_dTzE · 2023-07-04

**Soundness:** 4 excellent
**Presentation:** 3 good
**Contribution:** 2 fair
**Rating:** 7
**Confidence:** 3

**Summary:**

This paper proposes new convergence results for single-call stochastic extragradient methods under weaker conditions. More specifically, the authors consider quasi-strongly monotone and weak Minty VI problems, both under an unconstrained finite-sum (or arbitrary sampling) setting. The authors propose the expected residual (ER) condition, which extends similar conditions used in stochastic optimization (minimization) to the VI setting. ER is more general than the boundedness assumption on operator noise, and is sufficient for establishing their convergence results. The authors also give sufficient conditions for ER and explain its connections to other widely used technical conditions/assumptions. Using ER, the authors establish convergence results for single-call stochastic extragradient methods for quasi-strongly monotone and weak Minty VI problems. For quasi-strongly monotone problems, two results are given for constant and decreasing step size rules. The authors then give expressions for the ER parameters (which factor into convergence rates) under non-uniform sampling. Numerical experiments complement the theoretical findings.

**Strengths:**

- Clear presentation of results.
- Sufficient discussion of the background and context of the new ER condition.
- First convergence guarantee for SOG under arbitrary sampling.
- First convergence guarantee for SPEG without any bounded variance assumption.

**Weaknesses:**

See **Questions**.

**Questions:**

My main question is around the practicality of the proposed step size rules. In previous work, most (static, non-adaptive) step sizes used in convergence theorems tend to be highly conservative, and more aggressive or adaptive step size rules (which may weaken/break convergence guarantees) need to be used in order to make the algorithms converge in experiments.

In this work, to achieve desirable convergence rates, the step sizes (constant or decreasing) depend on $\delta$, $\mu$, and $L$. My concern is that (i) these constants may be hard to find, or (ii) only conservative, safe estimates can be given. Therefore, the step sizes given by the theorems may be too small. It seems that in all experiments, theoretically safe constants are used. Have the authors considered using more aggressive step sizes which do not follow the theorems entirely (but partially, such as the $1/k^2$ decreasing trend in (11) but with different constants)? If so, it would be helpful to point these out. If the theorems' suggested step size rules are already "near-optimal" in experiments (meaning that further increasing them would lead to non-convergence), it would also be worth pointing out this favorable observation. In fact, if true, this would be such a rarity based on my experience and other work, and should be worth highlighting.

**Limitations:**

This is a theoretical and methodological work. The limitations are mainly technical: many practical problems mentioned as a motivation of the work do not (or cannot be easily shown to) satisfy the exact technical assumptions. However, this is typical and not a concern.

I do not see potential negative societal impact.

---

> ### Author Rebuttal · Authors · 2023-08-09
>
> We thank the reviewer for a detailed review and positive evaluation. Below, we address questions and concerns raised by the reviewer.
>
> **\[My main question is around the practicality of the proposed step size rules...\]** We appreciate the reviewer's concern regarding the adaptive stepsizes. However, this should not be noted as a weakness of our work. In our work, we focus on the situation when the problem parameters are known since it is important to understand this case first before moving to the adaptive stepsizes. Not all papers on optimization should be about adaptive stepsizes, and it is beyond the scope of our work. There are various practical examples where the constants $L, \mu, \delta$ can be computed. One such example would be the Robust Least Square problem. Check out the Robust Least Square subsection under Numerical Experiments of the paper [Global Convergence and Variance Reduction for a
> Class of Nonconvex-Nonconcave Minimax Problems](https://proceedings.neurips.cc/paper/2020/file/0cc6928e741d75e7a92396317522069e-Paper.pdf). The objective function of equation 8 in this paper is a quadratic game, and the values of constants $L, \mu$ and $\delta$ can be easily computed.
>
> **\[...Therefore, the step sizes given by the theorems may be too small...\]** The reviewer writes the step sizes given by Theorems may be too small. However, we want to mention the fact that our analysis of SPEG recovers the best rate of convergence in a deterministic setting for both quasi-strongly monotone and weak minty variational inequality problems. This highlights the tightness of our stepsize choices.
>
> **\[... Have the authors considered using more aggressive step sizes which do not follow the theorems entirely (but partially, such as the $\frac{1}{k^2}$ decreasing trend in (11) but with different constants)?...\]** We did not try the stepsize choice of $\frac{1}{k^2}$ with a different constant. In the paper, we run experiments only to validate our theory. We appreciate this feedback and will add more details on this in the updated version of our work.
>
> **If you agree that we managed to address all issues, please consider raising your score. If you believe this is not the case, please let us know so that we have a chance to respond.**

---

> > ### Comment · Reviewer_dTzE · 2023-08-21
> >
> > Not all papers on optimization should be about adaptive stepsizes, and it is beyond the scope of our work.
> > - I agree that the limited scope of the paper makes sense, and it is not a fair ask to expand it during a short review window. I think a limitation rather than weakness is a better description.
> >
> > There are various practical examples where the constants can be computed.
> > - I partially agree to a smaller extend, given the RLS example the authors shared and some other highly stylized examples I thought about. Still, I don't think this is a good example of what's used "in practice" (we do mean different things here), and I believe that more aggressive step sizes will still help a lot for the RLS numerically based on similar problem/experiment settings I tried. I don't know a single example of mature code (optimization solvers, well-maintained open-source code, proprietary code used in the industry, in various application domains such as model training, forecasting, revenue optimization) where the step size follows, even remotely, the proposals in a principled methodological paper with convergence results. In short, almost every theory-grounded step size rule is too conservative and one should use much larger step sizes to boost numerical convergence - this is true in interior-point method solvers, SOTA extensive-form game equilibrium computation code, sparse SVM software, and almost all "real" optimization code I looked at. In these settings, theory-grounded step sizes can indeed be computed - but given its usually way too conservative and merely computing them takes nontrivial time, it's rarely done as such in these solvers/code. (DL training is a different topic and I don't think we want to go there...) Of course, this is a highly subjective opinion of my own based on my own experiences, and therefore it is not fair either to base my rating heavily on this. And it is also beyond the scope of this paper as the authors argued.

---

### Official Review · Reviewer_bysm · 2023-07-07

**Soundness:** 4 excellent
**Presentation:** 4 excellent
**Contribution:** 3 good
**Rating:** 6
**Confidence:** 3

**Summary:**

This paper studies the single-call stochastic extragradient methods for solving two classes of structured variational inequality (VI) problems, i.e., (i) quasi-strongly monotone problems and (ii) weak Minty variational problems. These two classes generalize the assumptions of strong monotonicity and comonotonicity to be only satisfied w.r.t. the solution point, respectively. The authors consider the stochastic reformulation of VIs, which allows for mini-batching with arbitrary sampling. The convergence results are built upon the expected residue condition, which can be explicitly implied by the component Lipschitzness and can imply bounded variance. Convergence are analyzed in both settings and with different stepsize strategies. The authors also provide numerical experiments to support their theoretical discussion.

**Strengths:**

1. The authors generalize the idea from stochastic minimization and propose the expected residue condition, which can be explicitly implied by the Lipschitzness of the component operator, and can be used to provide a variance bound. Thus, this paper does not require a bounded variance assumption nor growth conditions.
2. The authors provide a thorough discussion for the convergence results and detailed comparison with prior works, such as Lines 183--200.

**Weaknesses:**

1. Although the authors state the stochastic problem in term of the **finite-sum** structure (see Eq. 1 and 5), the convergence analysis is more likely for **infinite-sum** problems, especially when the authors also consider mini-batch in this paper.
2. The expected residue condition proposed in the paper looks restricted because the authors make this assumption directly on the stochastic estimator $g(x)$ instead of the stochastic oracle queries. It seems non-trivial to assume it for the stochastic estimators beyond the mini-batching ones, such as SVRG or SARAH.


**Questions:**

1. Although the weak Minty variational problems look more general than comonotonicity assumption, but I am not sure whether it is more of a theoretical artifact. Could the authors provide some practical examples which satisfy Def 1.2 but are not comonotone?
2. Could the authors clarify the stochastic settings this paper focused on, finite-sum or infinite-sum? I feel confused because $n$ never appears in the convergence results, and the choice of the batch size $\tau$ in Theorem 4.5 (Eq. 15) can essentially be larger than $n$.

**Limitations:**

Please see the weaknesses and questions above.

---

> ### Author Rebuttal · Authors · 2023-08-09
>
> We thank the reviewer for a detailed review and positive evaluation. Below, we address questions and concerns raised by the reviewer.
>
> **\[...the convergence analysis is more likely for infinite-sum problems, especially when the authors also consider mini-batch in this paper.\]** We consider only the finite-sum structure of the operator in our work. We have rigorous proofs that hold as long as (ER) condition holds. Indeed, it can be satisfied not only for finite-sum problems. However, it also works for the finite-sum case since we show that (ER) holds for finite-sum problems. Could the reviewer point us to the steps where the analysis does not work for finite-sum cases?
>
> **\[...non-trivial to assume it for the stochastic estimators beyond the mini-batching ones, such as SVRG or SARAH.\]** We make Assumption 3.1 directly about $g(x) = F_v(x)$. One can think about $g(x)$ as about oracle call or some other estimator constructed using some procedure (e.g., batching): our analysis holds whenever (ER) is satisfied. We provide multiple examples when (ER) holds. This assumption is not that restrictive as standard bounded variance assumption used in prior works in SEG.
>
>  Yes, our work does not capture the variance-reduced algorithms. However, this is not a weakness of our work. We provide the analysis without the bounded variance assumption, and our analysis captures several sampling strategies, including minibatch and importance sampling. However, we will try to have a unified study that will capture variance-reduced algorithms as the future work direction.
>
> **\[...Could the authors provide some practical examples which satisfy Def 1.2 but are not comonotone?\]** We don't have such a practical example in mind. However, we want to highlight that any comonotone problem is a special case of our Def 1.2. Therefore our convergence guarantee holds for comonotone problems as well.
>
> **\[Could the authors clarify the stochastic settings this paper focused on, finite-sum or infinite-sum?...\]** In our work, we focus on the finite-sum problems. However, if the assumptions (in particular, Assumption 3.1) are satisfied, our analysis works for different types of problems as well, e.g., when $F(x) = E_{\xi \sim \mathcal{D}}[F_{\xi}(x)]$ where $\mathcal{D}$ can be continuous distribution.
>
> **If you agree that we managed to address all issues, please consider raising your score. If you believe this is not the case, please let us know so that we have a chance to respond.**

---

> > ### Comment · Reviewer_bysm · 2023-08-13
> >
> > I thank the authors for their responses, which partially address my concerns. My score stands.
> >
> > However, the authors did not directly answer my Question 2 where I have doubts about the $O(K)$ batchsizes in Eq. (15), and the authors do not state the stochastic oracle complexity in the main paper. For the complexity, the authors implicitly assume that $n = \Omega(\frac{1}{\epsilon^2})$ to obtain an $\epsilon$-norm solution, which is kind of reduced to the infinite-sum setting.  For the other regimes $n = O(\frac{1}{\epsilon^2})$ the result in this paper is loose once the batch size in Eq. (15) is larger than $n$. For the paper [Böhm 2022] the authors compared to in Line 233, they assume the infinite-sum setting and provide the stochastic oracle complexity results. I believe the author should clarify these points in the revision.

---

> > > ### Author Response · Authors · 2023-08-15
> > > **Clarification on the oracle complexity**
> > >
> > > We thank the reviewer for the response.
> > >
> > > In Theorem 4.5, we use with-replacement batching meaning. Therefore, technically, this result allows the case of $n < K$. As we mentioned in our response, the proofs also work for the expectation case (infinite-sum).
> > >
> > > To get the complexity we need to choose $C = \frac{48}{\omega\gamma(1-L(\gamma + 4\omega))}$, $K = O\left(\frac{C|| x_0 - x^{\ast} ||^2}{\epsilon}\right)$ and $\tau = \max\left\lbrace 1, \frac{32 \delta}{(1 - L \gamma) L^3 \omega}, \frac{48 C \omega \gamma \delta || x_0 - x^{\ast}||^2}{(1 - \gamma L)^2 \epsilon}, \frac{2 C \omega \gamma \sigma_\ast^2 }{(1 - L \gamma) \epsilon} \right\rbrace$ and the oracle complexity will be
> > >
> > > $$K\tau = O\left(\max\left\lbrace \frac{C|| x_0 - x^{\ast} ||^2}{\epsilon}, \frac{32 C \delta || x_0 - x^{\ast} ||^2}{(1 - L \gamma) L^3 \omega \epsilon}, \frac{48 C^2 \omega \gamma \delta || x_0 - x^{\ast}||^4}{(1 - \gamma L)^2 \epsilon^2}, \frac{2 C^2 \omega \gamma \sigma_\ast^2 || x_0 - x^{\ast}||^2}{(1 - L \gamma) \epsilon^2} \right\rbrace\right).$$
> > >
> > > In particular, one can choose $\gamma = \max\lbrace \rho, \frac{1}{4L} \rbrace + \frac{1}{2L}$ and $\omega = \frac{1}{2}\min\lbrace \gamma - 2\rho, \frac{1}{4L} - \frac{\gamma}{4} \rbrace$.
> > >
> > > We will add the remark about the infinite-sum case and also the complexity bounds to the final version of our paper.

---

> > > > ### Comment · Reviewer_bysm · 2023-08-17
> > > >
> > > > I thank the authors' response, and I am happy with their answers.

---

> > > > > ### Author Response · Authors · 2023-08-21
> > > > > **Thanks**
> > > > >
> > > > > We thank again reviewer bysm for the positive review and the engagement during the discussion period. We also thank them for their point on the infinite-sum case and commit to adding further details on that, as explained above, to the final version of our paper.
> > > > > If you agree that we addressed all issues, please consider raising your mark to Accept (7).

---

### Official Review · Reviewer_rrNt · 2023-07-07

**Soundness:** 3 good
**Presentation:** 4 excellent
**Contribution:** 2 fair
**Rating:** 3
**Confidence:** 5

**Summary:**

This work studies single-call stochastic extra-gradient method for quasi strongly monotone and weak Minty Variational Inequality (VI). They relax the commonly-used bounded noise variance assumption and used the expected residual condition.

**Strengths:**

The paper is well-written and the problem is relevant.

**Weaknesses:**

In short, I think the technical novelty in this paper is minimal. I substantiate my claim below.


1. It introduces expected residual condition which has been studied before in SGD literature as the authors correctly state. So the condition itself is not a contribution. **As the authors agree (line 173-175), the main difference between this work and [29] is that (8) is an Assumption in [29] whereas Assumption 3.1 implies (8) in this work. But it is easy to see that the proof of Lemma 3.2 just requires a single use of Young's inequality (and indeed, that's how it's done in the Appendix).**

2. **Authors in [29] use another condition** $ E || g(x) − F(x) ||^2 \leq (a||x − x^*|| + b)^2 $ **which is extremely similar to ER condition. But in line $171$, the authors claim that the constants $a$ and $b$ are not available in closed form. But this is a trivial result (Proposition 3.3). These calculations are routinely done in literature involving  stochastic gradient as the authors themselves agree (line 153, [24-25]).**

3. **Use of ER condition relaxes the bounded condition on the noise variance but the additional technical innovation needed, compared to the bounded noise variance, is minimal.** Familiarity with the stochastic extra-gradient proof techniques right away reveals that the RHS of ER condition, $||x-x^*||_2^2$, is designed to be subsumed in other terms appearing in the proof which effectively leaves the proof almost same as the bounded variance case.

Specifically, **note that ER condition only introduces the term** $\omega^2\delta || \hat{x}_k - x^* ||_2^2$ **in the 5th line of the display appearing after line 664. But there is another term** $ -2 \omega \mu ||\hat{x}_k-x^*||_2^2$. **So the convergence can be guaranteed if** $\omega $ **is chosen small enough such that** $2\omega\mu ||\hat{x}_k-x^*||_2^2> \omega^2\delta ||\hat{x}_k-x^*||_2^2$. **This is almost trivial.**

In fact, one can easily think of allowing for bigger bounds for the noise variance keeping the proof unimpacted. For example, convergence can be easily shown if one allows for a noise variance bound like $\omega^{-1/2}\delta ||\hat{x}_k-x^*||_2^2$. **All the proofs depend on this simple and trivial extension.**

**Questions:**

Please see the weakness section. If you could elaborate on the novelty that would be great.

---

> ### Author Rebuttal · Authors · 2023-08-09
>
> We thank the reviewer for a detailed review. Below, we address questions and concerns raised by the reviewer.
>
> **\[...the condition itself is not a contribution\]** Here, we want to highlight that the ER condition in minimization literature involves the functional value, i.e. the right-hand side is $f(x) - f^\ast$ (check Assumption 3.1 [SGD for Structured Nonconvex Functions: Learning Rates,
> Minibatching and Interpolation](http://proceedings.mlr.press/v130/gower21a/gower21a.pdf)). However, there is no notion of function values in VIPs. Therefore, we have replaced the $f(x) - f^\ast$ with $||x - x^\ast||^2$, which completely changes the proof technique compared to the minimization setup.
>
> **\[...the main difference between this work and [29] is that (8) is an Assumption in [29]...\]** Yes, authors of [29](https://arxiv.org/pdf/2003.10162.pdf) also use similar conditions as we have mentioned in our work. But the reviewer ignores the setup considered in their paper. [29](https://arxiv.org/pdf/2003.10162.pdf) considers the extragradient method with two oracle calls and solves for operators satisfying error-bound condition, whereas we consider the single-call method for solving quasi-strongly monotone and weak minty variational inequalities. Even though the conditions are similar, our work is completely different in other aspects.
>
> **\[...These calculations are routinely done in literature involving stochastic gradient...\]** Yes, this kind of computation is not new; similar calculations exist in the literature, which we have also mentioned in our paper. However, we don't consider it a weakness of our work. We want to highlight that this was never done for the single-call extra gradient method for solving VIPs. As mentioned in the previous answer, our ER condition is a modification of the ER condition introduced in the minimization setup. Therefore, no work precisely computes the constants like $\delta$ of our ER. Moreover, we are the first to capture different sampling strategies (like minibatch, importance or any other single-element sampling) of the single-call method for solving VIPs.
>
> **\[...ER condition only introduces the term $\omega^2 \delta ||\hat{x_k} - x^\ast||^2$ in the 5th line of the display appearing after line 664...\]**
> - The reviewer rrNt argues ER condition only introduces the extra term $\omega^2 \delta ||\hat{x_k} - x^{\ast}||^2$, which can be handled easily. We politely disagree with the statement and ask the reviewer to check the 5th line after line 664 again. Reviewer rrNt completely ignores the term $\omega^2\delta||\hat{x_{k-1}} - x^\ast||^2$, which ER also introduced. Moreover, this term can not be canceled following the procedure mentioned by rrNt. Indeed, for handling such terms, we have introduced $||x_{k+1} - \hat{x}_k||$ in the Lyapunov function under consideration.
> - The reviewer also overlooks the different stepsizes proposed in our work. There were no switching stepsizes for single-call methods, even under the bounded variance assumption, to get exact convergence.
> - Moreover, reviewer rrNt also ignores our contributions for weak minty VIPs. We analyse stochastic single-call methods for $\rho < 1/2L$, improving the previous restriction on $\rho$.
>
> **We believe a score of 2 is too harsh for our work, and it definitely deserves better. If you agree that we addressed all issues, please consider raising your score. If you believe this is not the case, please let us know so that we have a chance to respond.**

---

> > ### Comment · Reviewer_rrNt · 2023-08-20
> > **Thanks for your answers.**
> >
> > Thank you for your detailed responses.
> >
> > **[...the condition itself is not a contribution]**  Similar assumption has been assumed in [29] as well. It is well known in saddle-point optimization literature (and VI literature) that function value-based assumptions or Lyapounov functions do not make sense and it has to be based on $||x\_t-x^*||_2^2$. The required changes in the proof techniques are extremely well-studied (see A2.b in [29], A 1.2. in [3] for example).
> >
> > **[...the main difference between this work and [29] is that (8) is an Assumption in [29]...]** [29]'s work is slightly different but my point is there is significant overlap that makes the novelty of this paper incremental.
> >
> > **[...These calculations are routinely done in literature involving stochastic gradient...]** I agree that the computation of $\delta$ was not explicitly done in previous literature. One reason for that is that the computation of $\delta$ is trivial given the finite-sum setup and requires basic algebraic manipulations.
> >
> > [...ER condition only introduces the term....in the 5th line of the display appearing after line 664...] These terms are easily controllable under ER condition. One main issue with Stochastic EG (SEG) is that with independent samples for the two steps of SEG, the algorithm will not converge even for monotone operators as proved in Proposition 1 of [29]. ER condition helps to alleviate this issue by providing a control over the variance proportional to the distance to the optimal point.
> >
> > **[misleading claims]** It is claimed in line 183-184 that "To the best of our knowledge, the above theorem is the first result on the convergence of SPEG that does not rely on the bounded variance assumption." This is simply not true. [29] has the unbounded variance assumption as well.
> >
> > I agree with the switching step-size and weak minty VIPs novelty.
> >
> > Overall: I agree that there are some novelties of the paper for which I am increasing my score by 1. But I still feel that the work (mainly the proof techniques required to establish the results) is too incremental to be considered for a venue like NeurIPS.
> >
> > [29] https://arxiv.org/pdf/2003.10162.pdf
> >
> > [3] https://arxiv.org/pdf/2111.08611.pdf

---

> > > ### Author Response · Authors · 2023-08-21
> > > **Response to rrNt**
> > >
> > > ### **Response to further comments:**
> > >
> > > **\[...the condition itself is not a contribution\]** We are not sure which part reviewer rrNt is referring to in A 1.2 of [3] and A 2.b. of [29].
> > >
> > > **\[...the main difference between this work and [29] is that (8) is an Assumption in [29]...\]** No, our work is not just slightly different from [29]. As mentioned in our earlier response, we solve a completely different class of problems (quasi-strongly monotone and weak minty) in the paper. Moreover, the algorithm analyzed in our paper uses single oracle calls in contrast to the two oracle calls used in [29]. Only the assumptions of stochastic estimators are closely related.
> > >
> > >
> > > **\[...ER condition only introduces the term....in the 5th line of the display appearing after line 664…\]** As mentioned earlier, analysis of the single-call method was done in the literature only under the bounded variance assumption. Our previous response also pointed out the difficulties of analyzing the SPEG. The analysis is not trivial. Moreover, we would like to emphaisize our contribution to solving weak minty problems for the reviewer once again. There does not exist any analysis of extragradient methods (both single and double oracle versions) without bounded variance assumption for solving weak minty problems. This work is the first to provide a convergence guarantee of extragradient methods without the bounded variance assumption for solving weak minty VIPs. We request the reviewer to check the proof techniques of weak Minty VIPs. We hope rrNt will recognize the difficulties and increase our score.
> > >
> > > **[misleading claims]** We disagree with the reviewer and stand by our claim. We ask the reviewer to check the algorithm in [29]. [29] analyses the stochastic extragradient method, which uses two oracle calls per iteration, while SPEG uses one oracle call per iteration. Therefore, as the paper claims, this is the first work that analyses the single-call extragradient method without bounded variance assumption, and our claim is **not misleading**.
> > >
> > > ------------------------------------------------------------------------------------------------------------
> > > ### **On novelty of our work:**
> > > Reviewer rrNt, mentions the following at the end of the review: “I still feel that the work (mainly the proof techniques required to establish the results) is too incremental to be considered for a venue like NeurIPS”.
> > >
> > > We respectfully stand by our claim of novelty (please check our full response) and we politely disagree with the reasoning of the above statement. In our opinion, the fact that part of the proof techniques is an extension of previous works is **no reason for suggesting rejection of a paper**.
> > >
> > > **With our work we answer several open questions in the performance of SPEG (see full response and also the last message to rev. Bwgk)  for solving structured non-monotone VIPs.**
> > >
> > > **We hope we have responded to the questions of reviewer rrNt appropriately. We still think a score of 3 is too low to recognize our contributions. We politely ask the reviewer rrNt to reevaluate our work.**

---

### Official Review · Reviewer_Bwgk · 2023-07-14

**Soundness:** 3 good
**Presentation:** 2 fair
**Contribution:** 2 fair
**Rating:** 6
**Confidence:** 5

**Summary:**

The paper explores single-call stochastic extragradient methods like stochastic past extragradient (SPEG) and stochastic optimistic gradient (SOG), which are increasingly popular and efficient algorithms for tackling large-scale min-max optimization and variational inequalities problems (VIP) commonly found in diverse machine learning tasks. Notwithstanding their growing popularity, the current convergence analyses of SPEG and SOG demand strong assumptions like bounded variance or growth conditions. Moreover, numerous key questions related to the convergence properties of these methods, such as mini-batching, effective step-size selection, and convergence guarantees under various sampling strategies, remain unaddressed.

This research endeavors to answer these questions, presenting convergence guarantees for two extensive classes of structured non-monotone VIPs: (i) quasi-strongly monotone problems, which extend strongly monotone problems, and (ii) weak Minty variational inequalities, which expand upon monotone and Minty VIPs. The paper introduces the expected residual condition and discusses its advantages, demonstrating how it enables a strictly weaker bound than those achieved by previously employed growth conditions, expected co-coercivity, or bounded variance assumptions. Lastly, the presented convergence analysis is applicable under the arbitrary sampling paradigm, encompassing special cases like importance sampling and various mini-batching strategies.


**Strengths:**

The main contributions of the paper include the following:

- **Expected Residual:** The authors introduce the Expected Residual (ER) condition for stochastic variational inequality problems. The ER condition is used to derive an upper bound on $E\|g(x)\|^2$ (as detailed in Lemma 3.2), offering a strictly weaker alternative to the bounded variance assumption and previously used "growth conditions" for the analysis of stochastic algorithms. The paper shows that the ER condition holds for a large class of operators, specifically when the $F_i$ of the problem are Lipschitz continuous.

- **Novel Convergence Guarantees:** The paper presents novel convergence guarantees for Stochastic Past Extragradient (SPEG) without the need for a bounded variance assumption in the cases of quasi-strongly monotone and weak Minty Variational Inequalities (MVI). This is achieved through the use of the proposed ER condition. For the class of quasi-strongly monotone Variational Inequalities Problems (VIPs), the paper demonstrates a linear convergence rate to a neighborhood of $x^*$ when constant step-sizes are utilized. Furthermore, theoretically motivated step-size switching rules are provided that guarantee exact convergence of SPEG to $x^*$. In the weak MVI case, the convergence of SPEG is proved for $\rho < 1/2L$, improving existing restrictions on $\rho$. The authors compare their results with the existing literature in Table 1.

- **Arbitrary Sampling:** By reformulating the variational inequality problem stochastically, the authors explain how their convergence guarantees for SPEG hold under the arbitrary sampling paradigm. This approach allows them to cover a wide range of sampling strategies for SPEG not previously considered, including mini-batching, uniform sampling, and importance sampling. Therefore, their analysis of SPEG is unified for different sampling strategies. The authors also demonstrate the tightness of their analysis by showing that the best-known convergence guarantees of deterministic Past Extragradient (PEG) for strongly monotone and weak MVI can be obtained as special cases of their main theorems.


**Weaknesses:**

- **Redundant Main Result:** The primary outcome of this research could essentially be seen as a single-call version of "Stochastic Extragradient: General Analysis and Improved Rates," but under simpler guarantees. This could potentially question the novelty of this research.

- **Uncertainty of Expected Residual (ER):** The Expected Residual (ER) is proposed as a new and beneficial condition, however, it is unclear whether this condition is indeed intuitive and interesting in practice. It would be beneficial if the authors could provide more justification or insights on the relevance of the ER condition in practical scenarios.

- **Importance of Single-Call Methods:** While the paper puts substantial focus on single-call methods, it is unclear whether these methods are significantly important in practical scenarios. Theoretical advantages are evident, but it would be constructive if the authors could provide more practical motivations or empirical examples where single-call methods bring notable improvements.

- **Comparison with Existing Methods:** If Stochastic Extragradient (SEG) methods already have satisfactory rates and guarantees, it is intuitive to assume that Stochastic Optimistic Gradient (SOG) methods would offer similar advantages. Therefore, a clear delineation between the contributions of the proposed method and existing ones would be beneficial for the reader.


**Questions:**

Address the weaknesses' section

---

> ### Author Rebuttal · Authors · 2023-08-09
>
> We thank the reviewer for a detailed review and positive evaluation. Below, we address questions and concerns raised by the reviewer.
>
> **\[... question the novelty of this research\]** We politely disagree with this remark. Our work on the single-call method uses a different proof technique compared to [Gorbunov et al. 2021](https://proceedings.mlr.press/v151/gorbunov22b/gorbunov22b.pdf) for proving the convergence. Analysing single-call methods involve dealing with the term $E||F_{v_k}(\hat{x_k}) - F_{v_{k-1}} (\hat{x_{k-1}})||^2$ (however, these do not appear in the analysis of the Extragradient method). Moreover, note that the Lyapunov function used to analyse our method is $E(||x_{k} - x^\ast ||^2 + ||x_{k} - \hat{x_{k-1}}||^2)$ in contrast to $E||x_{k} - x^\ast||^2$ for the Extragradient method. Moreover, we provide convergence guarantees for a larger class of non-monotone problems, i.e. weak Minty variational inequality, while Gorbunov et al. 2021 solve only quasi-strongly monotone problems. Therefore, our work is not just a simple extension of Gorbunov et al. 2021.
>
> **\[...provide more justification or insights on the relevance of the ER condition in practical scenarios.\]** As discussed in the paper, the Expected Residual is a more relaxed condition than the bounded variance assumption (bounded variance is used to provide convergence guarantees for the stochastic single-cell methods). Moreover, Expected Residual is not an assumption and holds for free whenever the operators $F_i$ are Lipschitz. However, there are several examples where bounded variance may not hold. We provide one such example here:<br>
> <br>Consider the simple linear regression problem: $$\min_{x \in \mathbb{R}} f(x) := \frac{1}{2} (a_1x - b_1)^2 + \frac{1}{2} (a_2x - b_2)^2$$ where $x \in \mathbb{R}$ and $f: \mathbb{R} \to \mathbb{R}$. Here let us denote $f_1(x)=(a_1x - b_1)^2$ and $f_2(x) = (a_2x - b_2)^2$. Now consider the estimator $g(x)$ of $\nabla f(x)$ under uniform sampling i.e. $g(x)$ takes the value $\nabla f_1(x)$ with probability $\frac{1}{2}$ and $\nabla f_2(x)$ with probability $\frac{1}{2}$. Then we have $$\mathbb{E} ||g(x) - \nabla f(x)||^2 = \frac{1}{2} ||\nabla f_1 (x) - \nabla f(x)||^2 + \frac{1}{2} ||\nabla f_2(x) - \nabla f(x)||^2 = \frac{1}{2} \cdot \frac{1}{4}||\nabla f_1 (x) - \nabla f_2(x)||^2 + \frac{1}{2} \cdot \frac{1}{4}||\nabla f_2 (x) - \nabla f_1(x)||^2 =  \frac{1}{4} ||\nabla f_1(x) - \nabla f_2(x)||^2 = \frac{1}{4} \left( 2(a_1x - b_1)a_1 - 2(a_2x - b_2)a_2 \right)^2 = \left( (a_1^2 - a_2^2) x - (a_1b_1 - a_2b_2) \right)^2.$$ Thus, the expression $\mathbb{E} ||g(x) - \nabla f(x)||^2$ is a quadratic function of $x$. Hence, as $x \to \infty$, we have $\mathbb{E} ||g(x) - \nabla f(x) ||^2 \to \infty$ and the variance can not be bounded by a constant. On the other hand, this function has a Lipscihtz gradient, and from our results, ER and condition (9) hold for free. In addition, an analysis under Expected Residual allows us to capture the performance of stochastic methods for various sampling strategies used in practice. For example, practitioners frequently use minibatch or importance sampling to train machine learning models. Our analysis with Expected Residual allows us to explicitly derive the convergence rates under different sampling strategies. However, bounded variance does not capture the analysis under such sampling strategies.
>
> **\[...it is unclear whether these methods are significantly important in practical scenarios...\]** The single-call extragradient method requires two oracle calls per iteration in contrast to the two oracle calls by the extragradient method. In order to make a fair comparison of the two methods, we should count the number of oracle calls required to achieve a given accuracy. We compared the stochastic extragradient method of [Gorbunov et al. 2021](https://proceedings.mlr.press/v151/gorbunov22b/gorbunov22b.pdf) with the stochastic past extragradient method. The number of oracle calls required by the single-call method was less than the extragradient method. We have provided a plot (.pdf file comparing SPEG and SSEG) to compare the two methods in th General Response to the reviewers. In the attached figure, the Stochastic Past extragradient requires 2000 oracle calls, while the stochastic extragradient method requires more than 3000 oracle calls for convergence. It highlights the lower computational complexity of the single-call method and provides empirical evidence of notable improvements for these methods.
>
> **\[...a clear delineation between the contributions of the proposed method and existing ones would be beneficial for the reader.\]**
> - As discussed in answer to the previous question, the empirical evidence shows that the single-call method can have lower computational complexity than the SEG methods. It motivates us to study the single-call methods in more detail.
> - We want to highlight that, previously, researchers used bounded variance to study single-call methods. However, we have provided an example in the previous answer of why this may not hold in simple scenarios. We lift such unrealistic assumptions and work under Expected Residual in our work.
> - Furthermore, we provide convergence guarantees of SPEG for various sampling strategies. We also provide empirical evidence to show the advantage of using importance sampling over uniform sampling for SPEG.
> - Moreover, we provide convergence guarantees of the stochastic single-call method for solving weak minty variational inequality problems with $\rho < \frac{1}{2L}$. It also improves the existing restriction on $\rho$ for stochastic methods. Previously, the best-known result used bounded variance assumption with $\rho < \frac{3}{8L}$ to provide a convergence guarantee of the stochastic method.
>
> **If you agree that we managed to address all issues, please consider raising your score. If you believe this is not the case, please let us know so that we have a chance to respond.**

---

> > ### Comment · Reviewer_Bwgk · 2023-08-20
> > **Answer**
> >
> > I have similar opinion with  Reviewer rrNt.
> > I don't have more questions. We will discuss later in the committee about the incremental or not of the contribution. Would you like to add about this aspect?

---

> > > ### Author Response · Authors · 2023-08-21
> > > **Final Comments**
> > >
> > > Thanks for the response. We want to mention that reviewer rrNt completely ignores our contribution to solving weak minty VIPs. Moreover, reviewer rrNt argues that the work of [29] is similar to ours. However, the ER condition and its variants are the only similarity between [29] and our work. The algorithm considered in our work (single-call version) and the classes of problems we solve differ entirely from what is considered in [29]. We have also mentioned the difficulties of analyzing the single-call method under ER conditions in our response to rrNt. The analysis of the single-call method under ER is more complex than rrNt states.
> > >
> > > For us it is clear that with our work we answer several open questions in the literature of solving VIPs and in particular for the performance of SPEG (one of the most popular algorithms in the literature). As a result our contribution can be significant for the ML community.
> > >
> > > Before our work the following questions were open:
> > > 1. What is the convergence performance of SPEG in the quasi-strongly monotone (3) and weak MVI (4) cases without assuming bounded variance assumption?
> > > 2. Can we relax the assumptions on stochastic estimators? (We use ER condition, which holds for free when the operators are Lipschitz.)
> > > 3. What are the beneficial step-size selection of SPEG in order to have faster practical performance for quasi strongly monotone and weak MVI VIPs?
> > > 4. Can we analyze SPEG beyond the classical uniform sampling? That is, can arbitrary sampling and importance sampling being used to improve the performance of the method and if yes how the convergence guarantees will capture that scenario?
> > > 5. Via numerical experiments we verify the tightness of our theoretical results. That is we show that the proposed step-size selections behave exactly as the theory suggested, we numerically evaluate the benefis of importance sampling (comapre to previou use uniform sampling) and we also show that our proposed theoretical results lead to faster convergence compared to the work [28].
> > >
> > > Having the above open questions in mind we believe that our work fully justifies acceptance to NeurIPS and we are opposite to the argument of rev rrNt that “feel that the work (mainly the proof techniques required to establish the results) is too incremental to be considered for a venue like NeurIPS”. We politely disagree with the reasoning of the above statement. In our opinion, the fact that part of the proof techniques is an extension of previous works is **NO reason for suggesting rejection of a paper**. The contributions of our work (answer major open questions to the area) are significant.
> > >
> > > We hope reviewer Bwgk will consider the abovementioned points while making the final decisions and even consider increasing their original score to support our work.

---

### Author Rebuttal · Authors · 2023-08-09

We thank the reviewers for their valuable feedback and time. In particular, we appreciate that the reviewers acknowledged the following strengths of our work:

- Reviewer rrNt acknowledges the problem (considered in our work) is relevant and needs to be addressed.
- All the reviewers identify relaxing assumptions like bounded variance or growth conditions as one of the main strengths of our work.
- Both Bwgk and dTzE recognize our work as the first to provide convergence analysis of single-call methods under arbitrary sampling.
- Reviewer Bwgk acknowledges the improvement provided in our work for solving weak minty VIPs (we improved the restriction on $\rho$ in the stochastic setting to provide convergence).
- All the reviewers think the paper is well-written, and we have done a good job presenting the prior literature and providing thorough discussion.
- Reviewers bysm and dTze appreciate our numerical experiments to support the theoretical findings.

The reviewers also have several questions and concerns that we address in our responses to each reviewer. We have attached a plot called SSEGvsSPEG.pdf here. This plot is for reviewer **Bwgk** to answer one of his questions. You can find more details about the plot in the response to **Bwgk**.

If the reviewers have further questions/concerns/comments, we will be happy to participate in the discussion.

---

### Decision · Program_Chairs · 2023-09-21

**Decision:**

Accept (poster)

**Comment:**

After carefully going through all the reviews, rebuttal, and the discussions, even though the technical contributions of the paper (e.g., introducing a new proof technique etc) are incremental, the overall contribution (broader problems, weaker assumptions, improved algorithms) is at the strong side. Hence I am recommending an acceptance for the submission.

Please implement and clarify all the changes requested by the reviewers.

That being said, while understanding that writing a rebuttal can be frustrating, I find the tone and the attitude of the authors quite aggressive and sometimes impolite. I suggest using a milder and more diplomatic tone in your next submissions and preferably not ask for specific scores from the reviewers.